# A single-cell analysis reveals tumor heterogeneity and immune environment of acral melanoma

Chao Zhang [1,7], Hongru Shen[2,7], Tielong Yang[1,7], Ting Li[1], Xinyue Liu[1], Jin Wang[1,3], Zhichao Liao[1], Junqiang Wei[1], Jia Lu[1], Haotian Liu [1], Lijie Xiang[1], Yichen Yang[2], Meng Yang[2], Duan Wang[4], Yang Li [2], Ruwei Xing[1], Sheng Teng[1], Jun Zhao[1], Yun Yang[1], Gang Zhao[5], Kexin Chen [6] ✉, Xiangchun Li[2] ✉ & Jilong Yang [1] ✉

Acral melanoma is a dismal subtype of melanoma occurring in glabrous acral skin, and has a higher incidence in East Asians. We perform single-cell RNA sequencing for 63,394 cells obtained from 5 acral and 3 cutaneous melanoma samples to investigate tumor heterogeneity and immune environment. We define 5 orthogonal functional cell clusters that are involved in *TGF-beta* signaling, Type I interferon, Wnt signaling, Cell cycle, and Cholesterol efflux signaling. Signatures of enriched *TGF-beta*, Type I interferon, and cholesterol efflux signaling are significantly associated with good prognosis of melanoma. Compared with cutaneous melanoma, acral melanoma samples have significantly severe immunosuppressive state including depletion of cytotoxic CD8+ T cells, enrichment of Treg cells, and exhausted CD8+ T cells. *PD1* and *TIM-3* have higher expression in the exhaustive CD8+ T cells of acral melanoma. Key findings are verified in two independent validation sets. This study contributes to our better understanding of acral melanoma.

Acral melanoma (AM) is a subtype of melanoma developed in the skin of the acral such as palm, sole, and subungual areas[1,2]. Although the absolute incidence of AM is rare, it accounts for approximately 50% of melanoma cases in East Asia. AM has a genetic landscape characterized by structural rearrangements and amplifications, and without the UV signatures of cutaneous melanoma (CM)[2]. For instance, AM has less common *BRAF* and *RAS* mutations and harbor a higher rate of *KIT* mutations and amplification. However, a phase II clinical trial showed that Imatinib was ineffective for AM with KIT mutations[3]. Recent studies showed that anti-PD1 treatment is less effective in AM[4]. While anti-

[1]Department of Bone and Soft Tissue Tumors, Tianjin's Clinical Research Center for Cancer, National Clinical Research Center for Cancer, Key Laboratory of Cancer Prevention and Therapy of Tianjin, Key Laboratory of Cancer Immunology and Biotherapy of Tianjin, Tianjin Medical University Cancer Institute and Hospital, Tianjin Medical University, 300060 Tianjin, China. [2]Tianjin Cancer Institute, Tianjin's Clinical Research Center for Cancer, National Clinical Research Center for Cancer, Key Laboratory of Cancer Prevention and Therapy of Tianjin, Key Laboratory of Cancer Immunology and Biotherapy of Tianjin, Tianjin Medical University Cancer Institute and Hospital, Tianjin Medical University, 300060 Tianjin, China. [3]Tianjin Academy of Traditional Chinese Medicine Affiliated hospital, 300120 Tianjin, China. [4]Department of Orthopedics, Orthopedic Research Institute, West China Hospital, Sichuan University, Chengdu 610041 Sichuan Province, China. [5]Department of Pathology, Tianjin's Clinical Research Center for Cancer, National Clinical Research Center for Cancer, Key Laboratory of Cancer Prevention and Therapy of Tianjin, Key Laboratory of Cancer Immunology and Biotherapy of Tianjin, Tianjin Medical University Cancer Institute and Hospital, Tianjin Medical University, 300060 Tianjin, China. [6]Department of Epidemiology and Biostatistics, Key Laboratory of Molecular Cancer Epidemiology of Tianjin, Tianjin's Clinical Research Center for Cancer, Key Laboratory of Cancer Immunology and Biotherapy of Tianjin, National Clinical Research Center for Cancer, Tianjin Medical University Cancer Institute and Hospital, Tianjin, China. [7]These authors contributed equally: Chao Zhang, Hongru Shen, Tielong Yang. ✉e-mail: chenkexin@tmu.edu.cn; lixiangchun2014@foxmail.com; yangjilong@tjmuch.com

PD-1 immunotherapy can increase the objective response rate of melanoma to 38%[5], but for AM it is only 16.6%[6]. Although the genomic characterization of AM has been intensively explored in several studies, it is still not clear why AM does not respond well to immunotherapy[7,8]. Therefore, it is needed to further explore the intrinsic features of AM from other aspects such as the tumor heterogeneity and immune environment by other more accurate methods, such as single-cell sequencing.

The ecological environment of melanoma composes of tumor cells, immune cells, fibroblast cells, and endothelial cells etc. Single-cell sequencing provides an unprecedented opportunity to dissecting tumor environment of melanoma. In recent years, a lot of studies were dedicated to exploring immune environment and tumor heterogeneity of melanoma. Tirosh and colleagues reported distinct features that had been linked to intrinsic resistance to *RAF/MEK* inhibition[9] in CM via single-cell sequencing. Andrade and colleagues reported two distinct gene expression programs of natural killer cells that are indicative of significant functional specialization such as cytotoxicity and chemokine synthesis in melanoma[10]. Sade-Feldman and colleagues identified two distinct states of CD8+ T cells associated with patient tumor regression or progression and a single transcription factor, *TCF7*, which could predict positive clinical outcome in an independent cohort of checkpoint-treated patients[11]. However, single-cell transcriptome profiling of AM remains unavailable. Therefore, we intended to explore microenvironment of AM.

In this study, we perform single-cell sequencing to systematically investigate the tumor heterogeneity and immune environment of AM and CM patients. We find that AM and CM patients are characterized by expression signatures of *TGF-beta* signaling, Type I interferon, Wnt signaling, Cell cycle, and Cholesterol efflux signaling. AM patients are featured by the severe immunosuppressive state in comparison with CM patients. The key findings are verified in two independent datasets.

## Results

### scRNA-Seq of the ecosystem of acral melanoma by deep learning

In the discovery set, we obtained 63,394 cells for 5 acral and 3 cutaneous tumor specimens from 6 melanoma patients subjected to scRNA-seq (Supplementary Fig. 1). Detailed clinical and pathological information were provided in Fig. 1A and Supplementary Table 1. The pathological diagnosis of all samples has been reconfirmed by pathologists. There are seven primary and one lymph node metastatic tumor samples. We collected the pre- and post-treatment samples from one patient who received immunotherapy (Supplementary Fig. 1). We performed scRNA-seq for 2 acral and 1 cutaneous tumor specimens collected from different time period as internal validation set and collected expression matrices of 9 acral tumor specimens conducted by Li and colleagues[12] as external validation set. Meanwhile, we performed whole-exome sequencing, immunofluorescence staining, drug treatment experiment at cellular level and transcriptome sequencing of 57 melanoma samples (Supplementary Fig. 1).

We identified 50 distinctive cell clusters (Fig. 1B) that belong to immune cells and non-immune cells by Miscell[13] (see Methods section). The immune cells were primarily divided into T cells (*CD3D, CD3E*), B cells (*MS4A1, CD79A*), natural killer cells (*FGFBP2, KLRD1*), monocytes, and macrophages (*LYZ, CD68, CD14*). The non-immune cell clusters were made up of melanoma cells (*MLANA, PMEL, MITF, DCT*), endothelial cells (*VWF, PECAM1*) and fibroblast cells (*COL1A1, COL3A1*) (Fig. 1C). These markers were able to distinguish cell types in the study conducted and defined by Tirosh and colleagues[9] (Supplementary Fig. 2A–C). We observed that AM and CM were distinguishable by the composition of these cell clusters, especially the immune cells (Fig. 1D). We observed that cells from the internal and external validation sets are well mixed with cells from the discovery set (Supplementary Figs. 3A and 4A with kBET coefficient of 0.846 and 0.827, respectively).

### Distinct functional signatures of the melanoma tumor cells

We grouped melanoma tumor cells into 5 main subgroups based on Gene Ontology analysis (see Methods section, Fig. 2A, B) and found that these 5 subgroups were characterized by distinct functional signatures (Signature1–5). Signature 1 was involved in cholesterol transportation and phospholipid efflux. Signature 2 was enriched for Wnt signaling pathway and oxidative phosphorylation circuits. Signature 3 was featured by enrichment of Cell cycle circuits such as G2M checkpoint and E2F targets. Signature 4 was associated with *TGF-β signaling*. Signature 5 was enriched for interferon response (Fig. 2C). These five signatures were discovered in the internal and external validation sets (Supplementary Figs. 3B–E and 4B–E) and exhibited high correlation with discovery set (Supplementary Figs. 3F and 4F).

### Pseudo-temporal transition trajectory of melanoma tumor cells

In pseudo-time analysis, we randomly selected 4980 high-quality tumor cells to establish a pseudo-temporal ordering reflective of cell lineage (see Methods section). Our result showed that the aforementioned 5 tumor cell subgroups were in different developmental states. Melanoma cells from Subgroups 2 and 3 were mainly at the root of phylogenetic tree. This might indicate that cell from Subgroups 2 and 3 were likely to be primitive tumor cells. Melanoma cells from Subgroups 1, 4, and 5 were at the mid-end of development with better differentiation (Fig. 2D). We also investigated the transcriptional changes associated with transitional states and observed that melanoma cells could be categorized into 3 pseudo-temporal phases. Phase 1 was predominated by Subgroups 2 and 3, which is characterized by upregulated genes expression of *UQCRH, PSMA7, LDHA,* and *NDUFC2* (Fig. 2E). Phase 2, predominated by Subgroup 1, was characterized by upregulation of *APOE, APOC1 and PLTP*. Phase 3 was dominated by Subgroups 4 and 5 upregulated by genes associated with Type I interferon signaling (*IFIT3, IFIT2, IRF1*) and heat response (*HSPA1A, HSPA1B*; Fig. 2E).

### Prognostic significance of melanoma tumor cell functional signatures

For the 6 patients enrolled for scRNA-seq in this study, they were divided into two groups C1/C2 according to their signatures and there is a trend of difference in overall survival between these two groups (Fig. 2F–H and Supplementary Table 1). We further examined the association of these signatures with survival in the TMCH-57 and TCGA SKCM cohort[14]. We clustered the 57 samples from TMCH-57 cohort into C1/2 groups. C1 is enriched for Signatures 2, 3 and C2 is enriched for Signatures 1, 4, 5 (Fig. 2I). The overall survival of C2 patients was significantly better than that of C1 (Fig. 2J, Log-rank test, HR = 0.34, 95% CI: 0.15–0.78, $p = 0.0075$). Overall survival of C1 versus C2 in AM remains different (Fig. 2K, Log-rank test, HR = 0.42, 95% CI: 0.18–1.02, $p = 0.05$). We divided 452 patients from TCGA skin cutaneous melanoma cohort into C1/2 groups based on enrichment score of the aforementioned five functional signatures (see Methods section). The C2 group was enriched for Signatures 1, 4, and 5, while the C1 group was enriched for Signatures 2 and 3 (Supplementary Fig. 5A). The C2 group has better overall survival outcome in comparison with C1 group (Supplementary Fig. 5B, Log-rank test, HR = 0.64, 95% CI: 0.46–0.88, $p = 0.005$).

In addition, we observed that these five expression signatures of AM were distinct from CM in our single-cell dataset (Supplementary Fig. 5C). For these 6 patients, 4 patients with AM patients were enriched for Signatures 1, 4, and 5 while the other 2 CM patients were enriched for Signature 2 and 3 (Supplementary Fig. 5D). And in the TMCH-57 cohort, almost all CM samples belonged to C1, while C2 basically consisted of AM samples (Supplementary Fig. 5E). The patients with AM had better overall survival than patients with CM (Supplementary Fig. 5F, Log-rank test, HR = 2.53, 95% CI: 1.27–5.05, $p = 0.007$). This speculation was further verified by survival analysis of

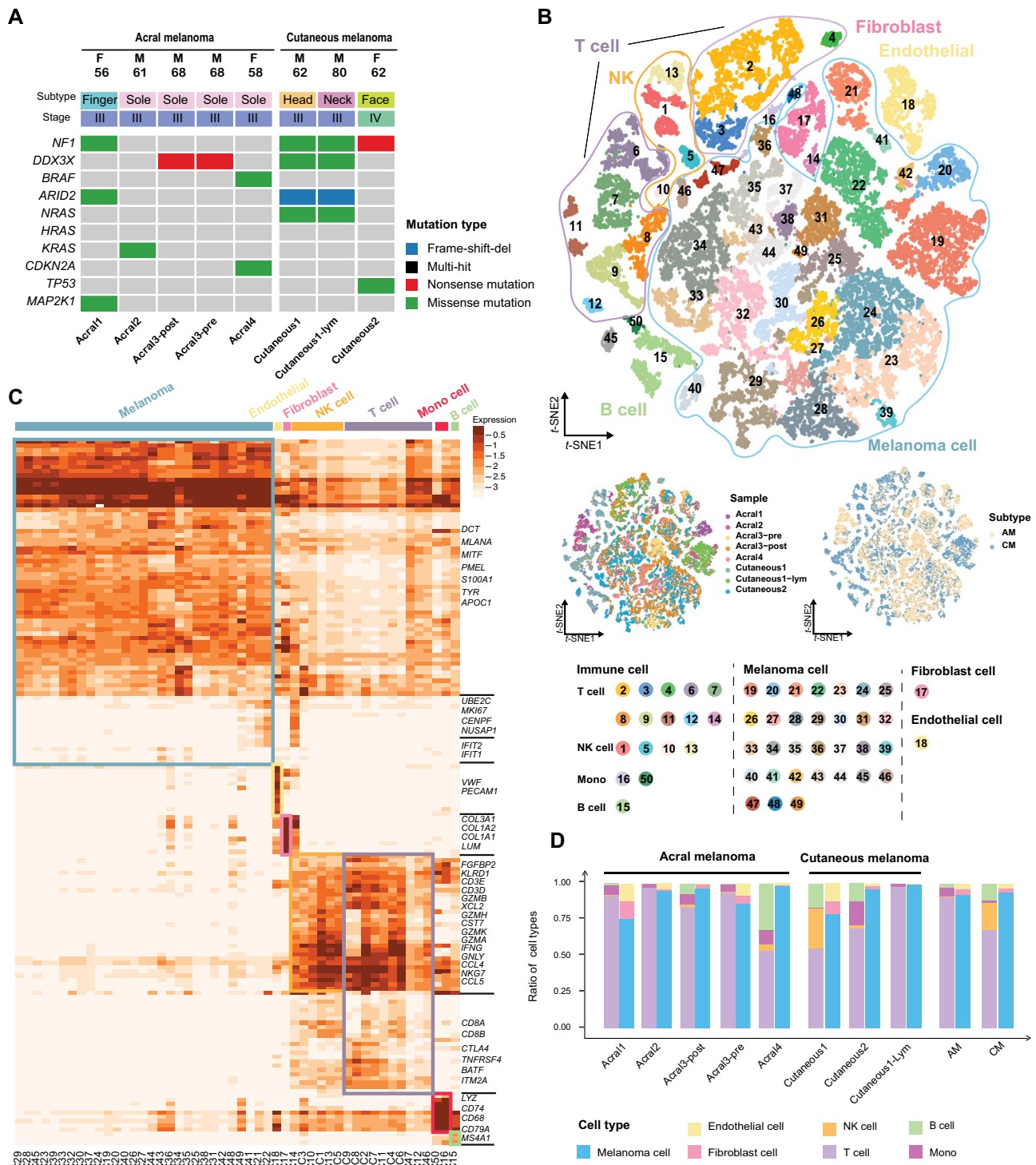

**Fig. 1 | scRNA-seq profiling of the acral and cutaneous melanoma environments.** **A** Landscape of driver mutations and basic clinical information of 8 samples underwent single-cell sequencing. (F, Female. M, Male.) **B** T-distributed stochastic neighbor embedding (t-SNE) plot, showing the annotation and color codes for cell types in the melanoma ecosystem and cell origins by color, patient origin (left panel), and AM or CM origin (right panel). (Mono, Monocyte) **C** Heatmap showing the expression of marker genes in the 50 cell clusters. The top bars label the clusters corresponding to specific cell types. **D** Histogram indicating the proportion of cells in tumor tissue of each analyzed patient. Non-immune cells and immune cells are shown in separate histograms. (Mono, Monocyte).

602 melanoma patients collected from Tianjin Cancer Hospital (Supplementary Fig. 5G, Log-rank test, HR = 1.86, 95% CI: 1.47–2.34, p = 9.18e-08). The survival of AM patients was better than CM in four clinical stages (Supplementary Fig. 5H–K). Therefore, we suggest that the difference in survival between AM and CM may originate from the different proportions of tumor cells with different functions.

## Immune microenvironment of acral melanoma

We divided the 16011 cells from cell clusters that were annotated to be immune cells (Fig. 1B) into 50 clusters and subsequently grouped them into 6 cell types (Fig. 3A, see Methods section). The identified cell clusters were featured by distinct marker genes (Fig. 3B). The T cell cluster consisted of 5 CD4+ T cell subgroups, 8 CD8+ T cell subgroups

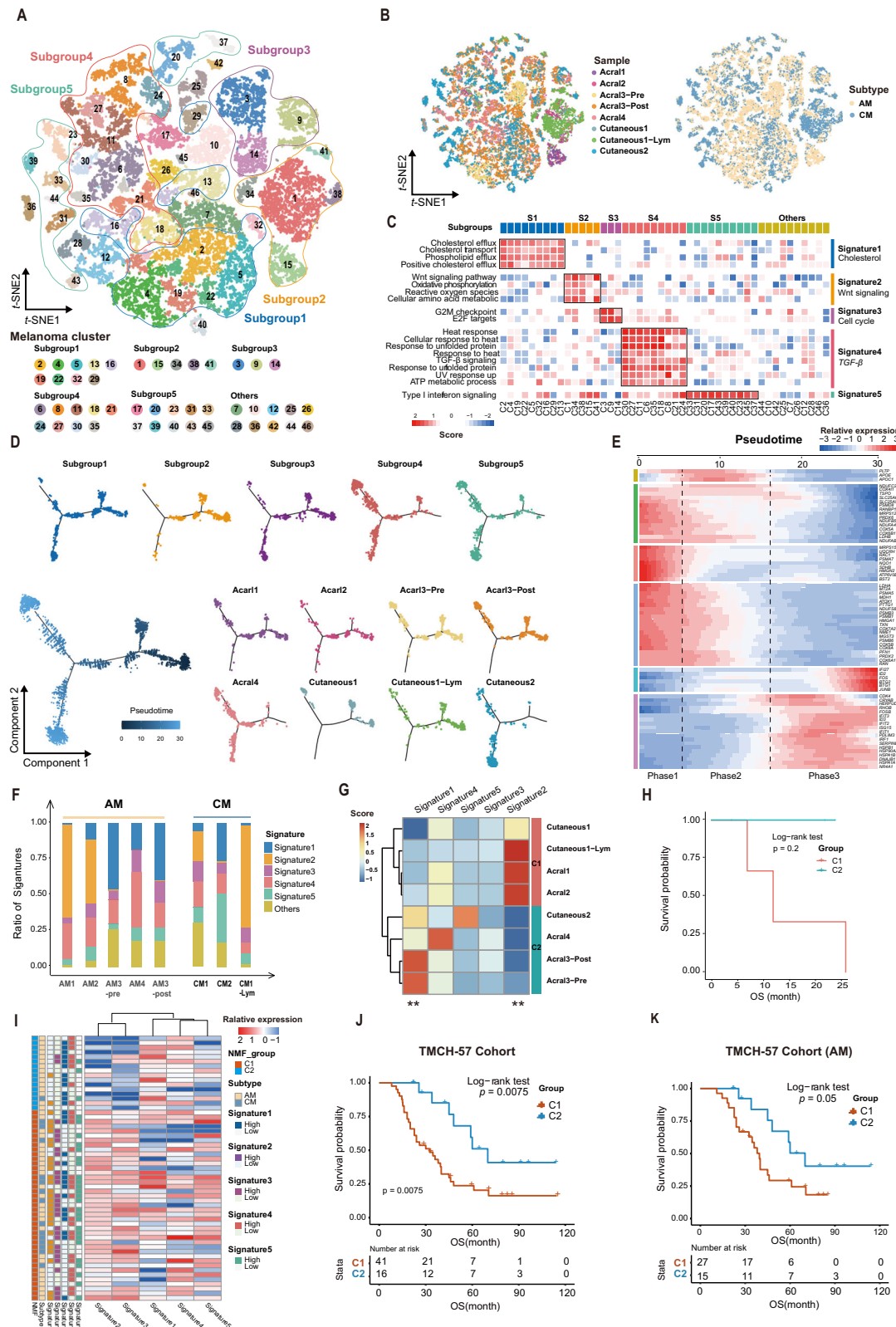

and Cell cycle T cells (Fig. 3C). The Treg subgroup highly expressed *IL2RA, FOXP3,* and *IKZF2,* co-stimulatory (*CD28, TNFRSF9,* and *ICOS*) and inhibitory markers (*TIGIT, CTLA4,* and *LAYN*). The CD4-CCR7, CD4-LEF1, and CD8-CCR7 subgroups were naïve T cells, which were marked with expression of *CCR7, LEF1,* and *SELL* genes. The CD4-NR4A1 and CD8-NR4A1 subgroups were tissue-resident memory T cells that were featured by high expression of *CD69* and *NR4A1*. The cytotoxic T cell

subgroups consisted of CD8-GZMK and CD8-MT1E that were char-acterized by high expression of *GZMK, GZMA, GNLY,* and *NKG7*, and low expression of genes in involved in immune checkpoint mediation such as *HAVCR2(TIM-3), PDCD1(PD-1)* and *LAG3*. CD8-PDCD1 and CD8-LAG3 subgroups were immunosuppressive CD8+ T cells that were marked with high expression of *CTLA4, HAVCR2 (TIM-3), PDCD1 (PD-1), LAG3,* and *TIGIT* (Fig. 3C).

**Fig. 2 | Five subgroups of melanoma cells. A** t-SNE plot showing the clusters and subgroups of melanoma cells. 5 subgroups are circled with corresponding colors. **B** t-SNE plot showing the cell origins by color, according to the samples (left panel) or subtypes (AM/CM, right panel). **C** Heatmap showing the expression score by ssGESA in the 46 clusters (5 Signatures), including biological functions and names of related signal pathways. Source data are provided as a Source Data file. **D** Pseudotime-ordered analysis of melanoma cells from AM and CM samples. Melanoma subgroups and samples are labeled by colors. **E** Heatmap showing the dynamic changes in gene expression along the pseudotime. **F** Histogram indicating the proportion of signatures in melanoma of each analyzed samples. **G** Heatmap showing the signature ratio of 8 single-cell samples and clustering into C1 and C2. Significance was determined using a two-sided, unpaired Wilcoxon rank-sum test relative to group C1 ($n$ = 4 samples) for group C2 ($n$ = 4 samples, Signature1 $P$-value = 0.029, Signature2 $P$-value = 0.029), **$P$-value < 0.05. **H** Kaplan−Meier analysis showing the overall survival rate of 6 patients, characterized by C1 (red) and C2 (green). **I** Heatmap showing the expression score of 57 TMCH-cohort patient's bulk-RNA data by ssGESA in the 5 signatures, including 2 clusters of NMF and 2 subtype of melanoma, and each signature categories into two groups with median value. Source data are provided as a Source Data file. **J** Kaplan-Meier analysis showing the overall survival rate of 57 TMCH-cohort patients, characterized by C1 (orange) and C2 (blue). The numbers of patients and the risk classification are indicated in the figure. Significance was calculated using the log-rank test. Source data are provided as a Source Data file. **K** Kaplan−Meier analysis showing the overall survival rate of 42 AM in TMCH-57 cohort, characterized by C1 (orange) and C2 (blue). The numbers of patients and the risk classification are indicated in the figure. Significance was calculated using the log-rank test. Source data are provided as a Source Data file.

We observed that marginal higher infiltration of Tregs in AM as compared with CM (Fig. 3D, median odds ratio = 7.42, adjusted $P$-value = 0.09). In the internal validation set, the proportion of Tregs in CM3 was significantly lower than in AM5 (Fisher's exact test, OR = 0.416, adjusted $P$-value = 1.246e-11) and AM6 (Fisher's exact test, OR = 0.393, adjusted $P$-value = 8.802e-12) (Supplementary Fig. 6B). Multiplex immunofluorescence staining of 8 samples (5 AM and 3 CM) confirmed that FOXP3+ Tregs is indeed more prevalent in AM than that in CM patients (Fig. 3E, F, Wilcoxon rank-sum test, Median: 0.137 versus 0.056, $P$-value = 2.337e-06). None of the other T cell subgroups showed significant differences in AM versus CM. In addition, CD8-MT1E, which is a CD8+ T cell subgroup, was characterized by high expression of *MT1E* and *MT2A* and enriched in two deceased AM patients (Supplementary Fig. 5L−N).

## CD8+ T cells reside in different transition trajectory states in acral melanoma versus cutaneous melanoma

We applied Monocle[15] to construct the developmental trajectories of the six aforementioned CD8+ T cell subgroups (see Methods section). The results showed that the pseudo-time trajectory initiates with CD8-CCR7 and CD8-NR4A1 subgroups via an intermediate state (i.e. CD8-GZMK and CD8-MT1E subgroups), and ends in the exhausted state (i.e. CD8-PDCD1 and CD8-LAG3 subgroups) (Fig. 4A). Starting with the naive signature, the cytotoxic signature continued from the middle of the trajectory towards the end, and the exhausted signature was predominantly upregulated at the end (Fig. 4B). Meanwhile, we found that the early and late states were mainly distributed in AM and lymph node metastases samples, while the intermediate state was predominated in CM (Fig. 4C). The transition distributions were different among acral, cutaneous and lymph node metastasis melanoma (Fig. 4C). As compared with CM, the exhausted signature score of AM was significantly higher (See Methods section; Fig. 4D, Wilcoxon rank-sum test, Median: 1.71 versus 1.39, adjusted $P$-value = 5.67e-05), while the cytotoxicity signature and resident scores were significantly lower (Fig. 4D, Wilcoxon rank-sum test, Median: 4.09 versus 4.88, adjusted $P$-value = 6.42e-08). In the internal validation set, the CD8+ T cells of AM samples exhibited lower cytotoxicity as compared with CM (Supplementary Fig. 6C, Wilcoxon rank-sum test, Median: 2.17 versus 4.30, adjusted $P$-value = 1.06e-12). We applied multiplex immunofluorescence staining to further confirm this result (see Methods section). As shown in Fig. 5A, B, samples from AM contained more exhausted CD8+ T cells (marked with PD1 and TIM-3), in contrast to CM samples, which were rich in cytotoxic CD8+ T cells (marked with GZMB).

These six CD8+ T cell subgroups were categorized into 3 phases based on transcriptional changes along developmental trajectories (Fig. 4E). The phase 1 was predominated by CD8-CCR7 and CD8-NR4A1 subgroups. Functional analysis showed that phase 1 was involved in positive regulation of lymphocyte activation, cellular response to heat, and cellular response to tumor necrosis factor (Fig. 4E). Phase 2 was characterized by the high expression of classical cytotoxic genes and low expression of T cell exhaustion markers. Phase 2 was involved in detoxification of copper ion (associated with *MT1E* and *MT2A*), antigen processing and presentation, and Wnt signaling pathway (Fig. 4E). Phase 3 was characterized by high levels of T cell exhaustion-related markers and associated with response to decreased oxygen levels (Fig. 4E). Among acral, cutaneous and lymph node metastasis melanoma, *GNLY* and *GZMA* appear to be up-regulated in Phase 2 but down-regulated in Phase 3 (Fig. 4F). *PRF1* was up-regulated from the beginning of the phase 2 towards the end of development, and were more prominent in AM (Fig. 4F). In the phase 3, *PD1* and *TIM-3* were highly expressed in AM (*PD1*, AM versus CM, $p < 0.01$; *TIM-3*, AM versus CM, $p < 0.01$), while *CTLA4, TIGIT*, and *LAG3* were more prominent in CM (Fig. 4G). The exhausted CD8+ T cells of the acral melanoma are also characterized by high expression of *PD1* and *TIM-3* in internal validation set (Supplementary Fig. 6D−F). Multiplex immunofluorescence staining experiments further confirmed that PD1 and TIM-3 were more expressed in AM, while GZMB was more prominently expressed in CM (Fig. 5A−C). Compared with CM, CD8+ T cells of AM were dominated by exhausted state and decreased cytotoxicity. These also indicated that AM patients are more likely to develop resistance to immunotherapy. In addition, the exhausted CD8+ T cells of AM patients were characterized by high expression of *PD1* and *TIM-3*. Therefore, selective multi-target immunotherapy may benefit AM patients.

## Combined anti-TIM-3 and anti-PD1 treatment increased tumor cell apoptosis in Acral melanoma

We performed in vitro cell experiments to further explore that combined application of anti-PD1 and anti-TIM-3 may benefit AM patients (see Methods section, Supplementary Fig. 1 and Fig. 5F−H). We isolated tumor cells and CD8+ T cells from fresh tumor tissues of AM and CM patients, respectively. CD8+ T cells were divided into 4 groups and co-raised with different immune checkpoint inhibitor drugs (including anti-PD1, anti-TIM-3, anti-PD1+ anti-TIM-3 and control groups) for 1 hour, and then co-cultured with tumor cells for 24 hours and detected the apoptotic ratio of tumor cells (Fig. 5F). The results showed that the apoptosis ratio of AM tumor cells were obviously increased in the anti-PD1 combined with anti-TIM-3 groups, and the combination group showed a superimposed effect. The apoptosis ratio of CM tumor cells was not obviously increased in the anti-TIM-3 group, and that of combination group was similar to that in the anti-PD1 group (Fig. 5G, H, gating strategy in Supplementary Fig. 10). Thus, our data suggest a strategy to overcome the resistance of immunotherapy in AM, such as the combination of anti-PD1 and anti-TIM-3.

## T cell repertoire of acral melanoma

To explore differences in T-cell receptors between the two melanoma subtypes, we performed immune repertoire sequencing on 6 of the samples involved in this study. The results showed that the number of

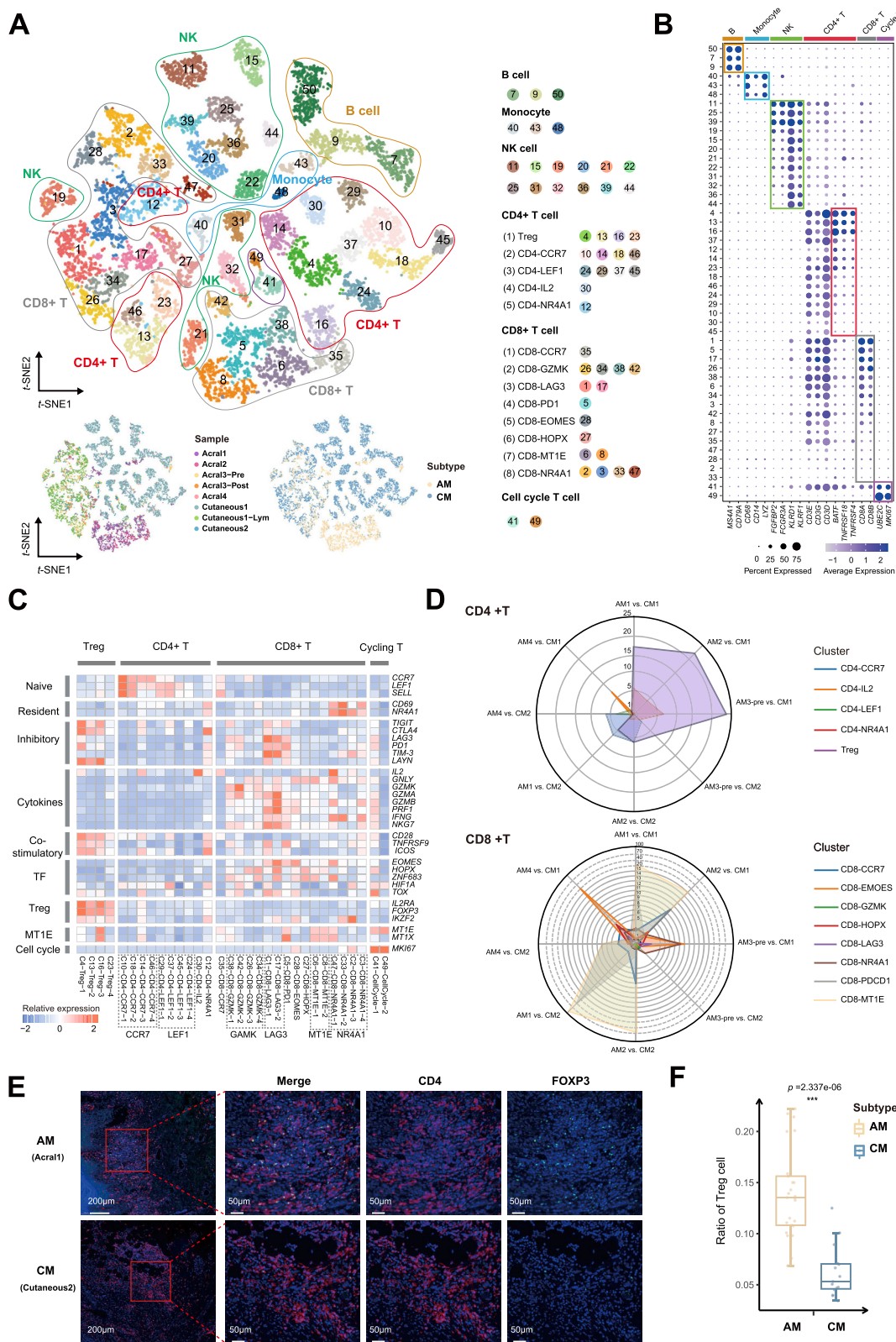

clonotypes presented a lower level in the AM samples compared to the CM samples (Supplementary Fig. 7A). We show the clonal proportion of each sample and the CDR3 length between the two melanoma subtypes in Supplementary Fig. 7B, C. Clonal diversity of T-cell receptors in AM also showed lower levels, compared with CM (Supplementary Fig. 7D–F). VDJ gene usage of T cell receptors has been shown in Supplementary Fig. 7G.

## Differences before and after treatment in an immune-resistant patient with acral melanoma

We compared the changes in the proportion of immune cells in the two AM samples (Acral3-pre and Acral3-post) before and after immunotherapy in one patient. It was found that the pre-treatment samples had more CD8+ T and Cycle T cells, while the post-treatment one was more prominent in CD4+ T and B cells (Fig. 6A). We also identified the

**Fig. 3 | Immune cell components in acral and cutaneous melanoma. A** t-SNE plot showing the clusters of immune cells and cell origins by color, according to immune cell types (upper panel), samples (lower left panel), and subtypes (AM/CM, lower right panel). **B** Dot plot showing percent expression and average expression of 50 immune cell clusters, including 6 main types of immune cells. The top bars label the clusters corresponding to specific cell types. **C** Heatmap indicating the expression of selected gene sets in T subtypes, including naive, resident, inhibitory, cytokines, co-stimulatory, transcriptional factors (TF), Treg, MT1E, cell cycle, and cell type. Source data are provided as a Source Data file. **D** Radar plots depicting the Odds Ratio between a pair of AM and CM samples. **E** Representative images of

multiplex immunofluorescence staining in formalin-fixed paraffin-embedded (FFPE) tissues, indicating CD4+ FOXP3+ cells, in paired AM and CM samples. Scale bar, 200 μm and 50 μm. **F** Boxplots illustrating the fraction of Treg in AM (yellow; Acral1, Acral2, Acral3-pre, Acral3-Post, Acral4) and CM (blue; Cutaneous1, Cutaneous2, Cutaneous1-lym), respectively. Box center lines, bounds of the box, and whiskers indicate medians, first and third quartiles, and minimum and maximum values within 1.5×IQR (interquartile range) of the box limits, respectively. Significance was determined using a two-sided, unpaired Wilcoxon rank-sum test relative to AM ($n = 25$ fields) for CM ($n = 15$ fields, P-value = 2.337e-06). Source data are provided as a Source Data file.

differentially expressed genes (DEGs) of CD8+ T cells between these two samples (Fig. 6B). The pathways of immune functions were upregulated in pre-treatment such as antigen processing and presentation and NF-kappa B signaling, while those pathways were not enriched in post-treatment sample (Fig. 6C). In addition, we compared the developmental trajectories of CD8+ T cells before and after the treatment. CD8+ T cells in the phase 2 of trajectory, cytotoxic CD8+ T cells, were higher abundance before the anti-PD1 therapy (Fig. 6D), and genes (*GNLY*, *GZMB* and *PRF1*) related to cytotoxicity also showed high expression levels (Fig. 6E and Supplementary Fig. 8A, B). The results of the multiplex immunofluorescence staining experiment further confirmed that the proportion of CD8+ T cells decreased, while the proportion of CD4+ T cells and B cells increased in the Acral3-post sample (Fig. 6F, G). To validate our investigation about the changes of the proportion of immune cells during the immunotherapy, two validated datasets of melanoma patients who had received immunotherapy were used[11,16]. In the Sade-Feldman's cohort[11], non-response patients showed a significant increase in the proportion of CD4+ T cells and a decrease trend in the proportion of CD8+ T cells (Supplementary Fig. 9A–C). Ratio of these immune cells in response group is provided in Supplementary Fig. 9D. In the 42 pairs of Bulk-RNA cohort[16,17], which patients with CM, the results also showed the same trend, the proportion of CD8+ T cells decreased in SD and PD groups (Supplementary Fig. 9E, F). Ratio of these immune cells in CR and PR group is provided in Supplementary Fig. 9G, H. The phenomenon indicated that the cytotoxicity of CD8+ T cells was reduced and the original immune-related functions were defuncted after receiving anti-PD1 therapy.

We also found that the genomic amplification of chromosome 4 was apparent in melanoma cells of the post-treatment sample but absent in those of pre-treatment sample (Supplementary Fig. 8C). There are 44 on chromosome 4 exhibited differential expression in cells from post-treatment sample as compared with pre-treatment sample (See Methods, Supplementary Fig. 8D). These 44 genes also exhibited higher expression in post-treatment sample versus pre-treatment CM samples in the PD group patients from the study of clinical trial (NCT01621490) dataset[16,17] (Supplementary Fig. 8D). Meanwhile, 12 of these genes were significantly overexpressed in PD patients compared to CR and PR patients (Supplementary Fig. 8E). These 44 genes were enriched in EGFR signaling and cell cycle phase transition (Supplementary Fig. 8F).

## Discussion

We provided a single-cell transcriptome landscape for AM in comparison with CM. Our study provided a broad understanding of tumor microenvironment and cellular composition in AM. We unveiled 5 functional signatures from melanoma cells and linked them to the prognosis. AM is different from CM in that AM patients are mainly composed of Signatures 1, 4, and 5, which are related to good prognosis. This finding was exemplified by the better overall survival of AM patients as compared with the CM. Study from Lim and colleagues also reported that the overall survival of AM was better than that of non-acral melanomas based on a single instiution study in Korea[18]. Thus, we suggest that the melanoma tumor cell heterogeneity which is

characterized by the different signatures might has prognostic significance, regardless of the tumor types of AM or CM.

In tumor immunity, Treg cells are involved in tumor development and progression by inhibiting antitumor immunity[19]. Treg cells induced by the *PD-1* pathway may also assist in maintaining immune homeostasis, keeping the threshold for T-cell activation high enough to safeguard against autoimmunity[20]. We found that Treg cells were higher in AM than those in CM. Higher abundance of Treg is associated with immunotherapeutic resistance. Thereby, we speculated that the response rate of AM is lower than CM in the context of immunotherapy. Recent study showed that the ORR of AM (16%) was lower than the ORR of no-acral CM (31%)[4,6]. An explanation to this observation is that infiltration of Treg prevent tumor cells from being killed by immune cells, and thus develop resistance to immune checkpoint inhibitors[21,22].

AM had a high proportion of CD8+ T cells in the initial state and exhausted state. CD8+ T cells with cytotoxicity signature were relatively few in AM and quickly transformed into an exhausted state of high expression immunosuppressive marker genes. Our result revealed that *PD1* and *TIM-3* were highly expressed in samples of AM, while *LAG3, TIGIT*, and *CTLA4* displayed lower expression. Those indicated that combination of *PDCD1 (PD-1)* and *HAVCR2 (TIM-3)* blockades might benefit patients with AM. This inference has also been further confirmed in drug experiments at the cell level in vitro. It is also expected to make further progress in animal experiments and random clinical trials. By contrast, patients with CM were characterized by exhausted CD8+ T cells that expressed high level of *LAG3* and *CTLA4*. This implied that the combination of *LAG3* and *CTLA4* inhibitors might be more appropriate for CM patients.

We observed that Immune infiltration is scarce in patients with AM, which was also reported in previous study[12]. However, AM has better overall survival as compared with CM. This is probably due to differences of tumor signatures underlying AM and CM. For instance, AM was enriched for cholesterol metabolism. Upregulation of the cholesterol metabolism was associated with favorable survival in lower grade glioma[23].

In the comparative analysis of samples before and after immunotherapy, it was showed that the patient's immune environment had undergone major changes. The post-treatment immune environment had more CD4+ T and B cells, while less CD8+ T and Cycle T cells comparing to the pre-treatment sample. Similar trend was validated in other datasets with the cutaneous melanoma patients received immunotherapy. The CD8+ T cells of the samples post-treatment had lost their original immune function, and the ratio of CD8+ T cells with cytotoxicity was reduced. These are probably the reasons why the anti-PD1 treatment of this AM patient was not effective.

Interestingly, we found genomic amplification of chromosome 4 in two AM samples and identified 44 differential expression genes which may be associated with immunotherapy resistance. These 44 genes were enriched in *EGFR* and cell cycle pathway. A previous study showed that *EGFR* can up-regulate the expression of *PD-L1*, and causing immunosuppression. This means that immunotherapy combined with *EGFR* pathway related gene (*TRIM2, GAB1, SPRY1*) inhibitors may improve the effectiveness of immunotherapy[24]. In addition, we found that 12 of these 44 genes exhibited higher expression in the SD and PD

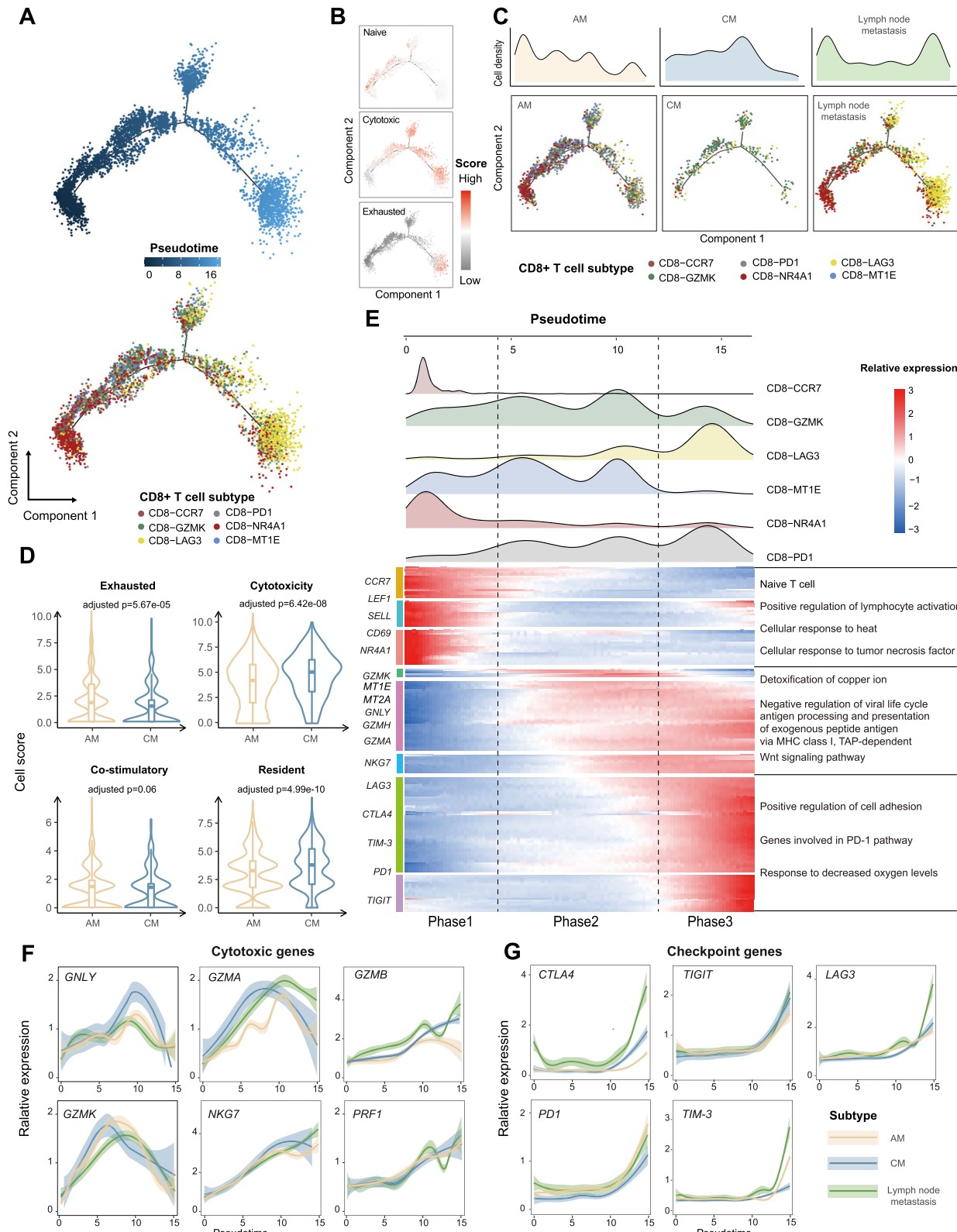

group in comparison with CR+ PR group patients with cutaneous melanoma. This finding needs to be further verified in the data set of acral melanoma patients receiving immunotherapy.

The melanoma samples subjected to single-cell sequencing are from Chinese patients and the sample size is limited. The key findings in our study was partially verified in the external dataset from melanoma patients treated in the US (race and ethnicity information not provided)[9,11,12]. However, it remains elusive with respect to the similairties and differences of tumor heterogeneity and microenvironment in melanomas among different races and ethnicities. Recently, Biermann and colleague dissected tumor ecosystem of treatment-naïve melanoma patients with brain metastasis[25]. It is also worthwhile to integrate and compare data from our study and Biermann and colleague to provide better understanding of racial disparities in melanomas.

**Fig. 4 | Analysis of CD8+ T cell transition states in acral melanoma and cutaneous melanoma samples. A** Pseudotime-ordered analysis of CD8+ T cells from AM and CM samples. T cell subtypes are labeled by colors. **B** 2D pseudotime plot showing the dynamics of naïve (upper panel), cytotoxic (middle panel), or exhausted signals (lower panel) in CD8+ T cells, from AM and CM samples. Source data are provided as a Source Data file. **C** 2D graph of the pseudotime-ordered CD8+ T cells, from AM (left panel), CM (middle panel), and Lym (right panel) samples. The cell density distribution, by state, is shown at the top of the figure. Source data are provided as a Source Data file. **D** Violin plot showing the expression of co-stimulatory, cytotoxic, resident, and inhibitory signature genes in CD8+ T cells in AM (yellow; Acral1, Acral2, Acral3-pre, Acral4) and CM (blue; Cutaneous1, Cutaneous2) samples. Box center lines, bounds of the box, and whiskers indicate medians, first and third quartiles, and minimum and maximum values within 1.5×IQR (interquartile range) of the box limits, respectively. Significance was determined using a two-sided, unpaired Wilcoxon rank-sum test relative to AM ($n = 2422$ cells) for CM ($n = 400$ cells, adjusted $P$-value were Exhausted, 5.67e-05, Cytotoxicity, 6.42e-08, Co-stimulatory, 0.06, Resident, 4.99e-10). Source data are provided as a Source Data file. **E** Heatmap showing the dynamic changes in gene expression along the pseudotime (lower panel). The distribution of CD8 subtypes during the transition (divided into 3 phases), along with the pseudo-time. Subtypes are labeled by colors (upper panel). **F, G** Two-dimensional plots showing the dynamic expression Cytokines genes (**F**) and checkpoint genes (**G**) during the T cell transitions along the pseudo-time. Error bands show local polynomial regression and the 95% confidence interval, respectively.

In conclusion, this study enables better understanding of the tumor ecosystem heterogeneity between AM and CM, in terms of immune and tumor phenotypes. Our results can be a valuable resource, facilitating a deeper understanding of the mechanisms associated with AM and assisting in developing more effective therapeutic targets and biomarkers for immunotherapies in AM patients.

## Methods

### Ethic approval
All clinical specimens in this study were collected with informed consent for research use and were approved by the Tianjin Medical University Cancer Hospital institutional Review Boards in accordance with the Declaration of Helsinki, under protocol number bc2022110. Consent to publish relevant clinical information potentially identifying individuals (e.g., age, gender, overall survival time, clinical stage, etc.) was obtained. No compensation was provided to the participants in this study.

### Single-cell RNA sample collection and sequencing
We collected 5 acral and 3 cutaneous melanoma samples as discovery set and 2 acral and 1 cutaneous melanoma samples as internal validation set. The tissue samples were obtained with patient informed consent and approval of the Tianjin Medical University Cancer Institute and Hospital. Fresh tumor samples were surgically removed from patients and immersed in a complete medium containing 90% Dulbecco's modified eagle medium (DMEM; catalog number: 11054001, GIBCO) and 10% fetal bovine serum (FBS; catalog number: 16140071, GIBCO), and transported to the lab in a refrigerated container. Suitable small tissue blocks were cut into pieces (diameter 1–3 mm). Single cells were prepared in the Chromium Single Cell Gene Expression Solution using the Chromium Single Cell 3′ Gel Bead, Chip, and Library Kits v2 (10× Genomics) as per the manufacturer's protocol. In all, about 10,000 total cells were loaded to each channel with an average recovery of 8000 cells. The cells were then partitioned into Gel Beads in Emulsion in the Chromium instrument, where cell lysis and barcoded reverse transcription of mRNA occurred, followed by amplification, shearing, and 3'adapter and sample index attachment. Libraries were sequenced on Hiseq Xten at BGI, Beijing, China. On average, 120 Gb of raw data were generated for each sample.

### Single-cell sequencing analysis
Raw base call (BCL) files were analyzed using CellRanger (v2.1.1). The "mkfastq" command was used to generate FASTQ files and the "count" command was used to generate raw gene-barcode matrices aligned to the 10X Genomics GRCh38 Ensembl build 84 genome (v1.2.0). The data from all 7 samples were combined in R (v3.6.2) using the Read10X function from the Seurat package (v3.1.5), and an aggregate gene expression matrix and Seurat object (63,394 samples and 35124 genes) were generated. The gene expression matrix was normalized by log2 transformation and scaled each gene by subtracting its mean and dividing with standard deviation. We extracted feature representations of single-cell using Miscell[13]. The gene expression signatures of single cells were captured by deep neural network. In this study, Momentum Contrast algorithm[26,27] was used to learn similar representation of single-cell expression by narrowing the gap between the augmented and corresponding original gene expression profiles, which consisted of a deep neural network of 63 layers with dense connection as feature encoder and multi-layer perceptron (MLP) as project head to map features learned by the encoder network to space where contrastive learning is applied. Data augmentation was used to increase data diversity and mimic data variation, and the operations included random shuffling or zeroing out 20% of gene expression values. Stochastic gradient descent algorithm[28] was used to train the model in parallel on two graphic processing units for 299 epochs with an initial learning rate of 0.24, weight decay of 0.0001 and batch size of 256. The learning rate was decayed by 0.1 at epoch 150 and 250. The model was developed with PyTorch package (v 2.3.1). The K-nearest neighbor graph was built on gene expression signatures of single-cells using Scanpy[29] (v2.1). The gene expression signatures of single-cell were embedded into two dimensions by t-distributed stochastic neighbor embedding[30] (t-SNE). The neighbor graph was used to find clusters by Leiden algorithm[31]. and a n-neighbors parameter set to "k = 5". The cluster-specific marker genes were identified by MAST algorithm[32] using Seurat "FindAllMarkers" function. The resulting single-cell clusters were visualized in t-SNE representations and annotated to biological cell types by canonical marker genes (Fig. 1C).

### RNA-sequencing data analysis
RNA sequencing was performed using frozen tissue extracts of 57 melanoma patients. RNA was extracted following the Trizol reagent (Invitrogen™, catalog number: 15596026) manual. mRNA library was constructed using MGIEasy mRNA library kit following the manufacturer's instructions. Libraries were sequenced on an DNBSEQ-G400 sequencer for PE150 cycles. Reads were aligned to a human reference genome (GRCh38) using subread aligner[33]. The RNA read counts were normalized using the edgeR R package. Expression of each RNA was transformed to TPM. Gene set enrichment analysis was performed using the GSVA package (v1.34.0) and plots with pheatmap package (v1.0.12) in R.

### Whole exome sequencing
We performed whole-exome sequencing (WES) for frozen tissues obtained from 8 patients with melanoma. We dissolved the frozen tissue (~10–12 8 μm sections) in PBS in order to remove OCT. Genomic DNA of the tissue was subjected to DNeasy Blood and Tissue Kit (QIAGEN, Hilden, Germany). We used the WES capture kit of Agilent V6 (60 Mbp) and performed pair-end sequencing of 100 bp for ×100 coverage on the BGISEQ-500 instrument (BGI Group, Shenzhen, China).

### TCR sequencing and analysis
We amplified immune repertoires of TCRs by amplicon-rescued multiplex nested PCR with primers targeting V- and C-genes (iRepertoire, Huntsville, AL, USA). Firstly, 1 μg of RNA was placed in a reaction with HTIvc (TCR) primers for different barcodes (iRepertoire) and reagents

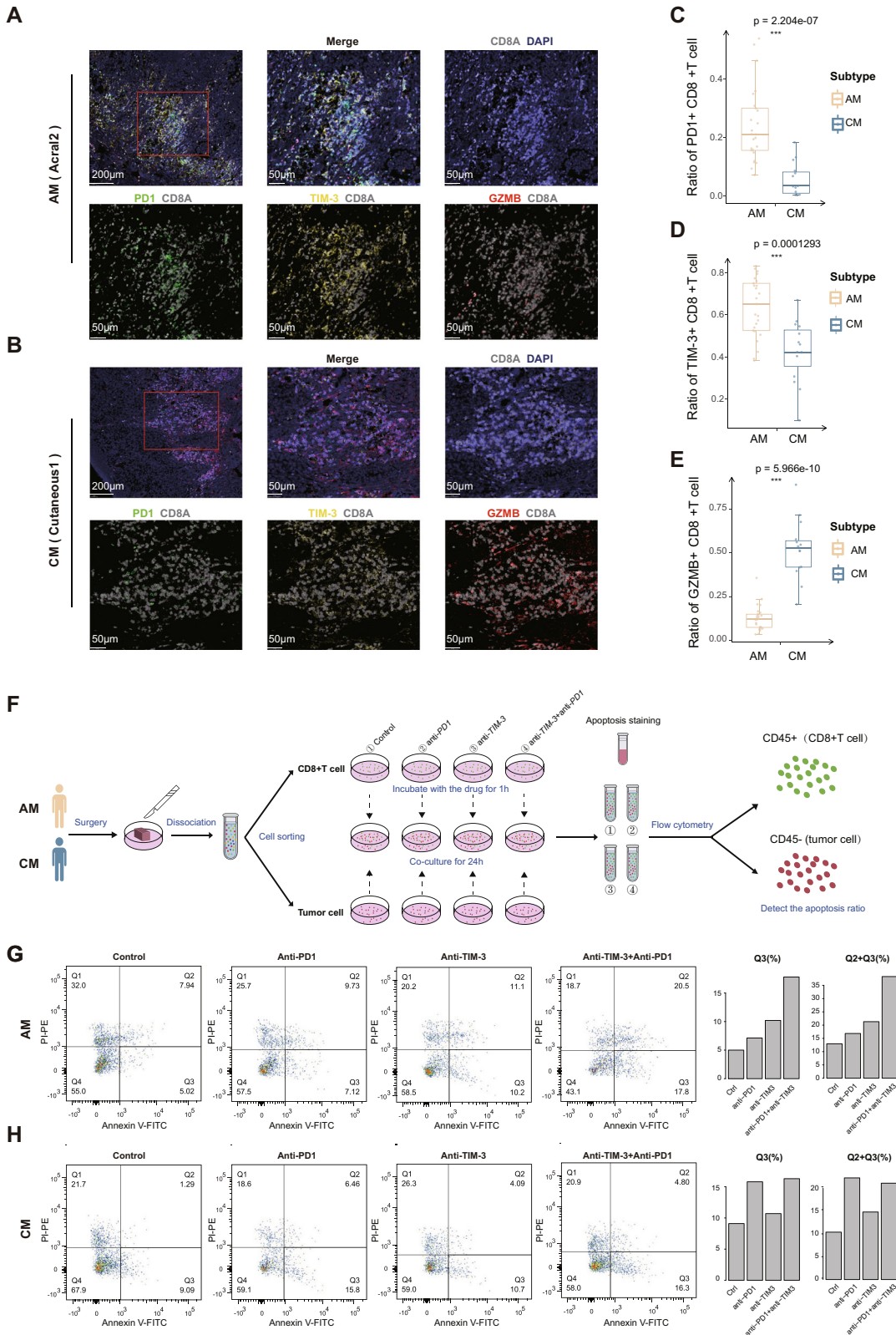

from the OneStep RT-PCR kit (Qiagen). Secondly, 2 μl of PCR1 products were placed in a reaction containing Illumina common primers (iRepertoire) and reagents from Multiplex Master Mix kit (Qiagen). The PCR2 products were purified using QIAquick gel extraction Kit (Qiagen). Sequencing was conducted on the BGISEQ-500 instrument (BGI Group, Shenzhen, China). Alignment onto the germline V-, D-, and J-genes was performed with Smith–Waterman algorithm. Identity was

called according to IMGT/GENE-DB[34]. R package "immunarch" (v 0.6.7) was used to analyze the data.

## Composition of acral melanoma

The AM tumor cell subgroups were clustered based the biological function, and the shared genes of these clusters were identified as the marker genes of melanoma tumor function modules. Single-sample

**Fig. 5 | Validation experiment of multiplex immunofluorescence staining and drug treatment at cellular level. A, B** Representative images of Multiplex immunofluorescence staining in Formalin-fixed paraffin-embedded (FFPE) tissues, indicating CD8A+ PD1+, CD8A+ TIM-3+, and CD8A+ GZMB+ cells, in paired AM (**A**) and CM (**B**) samples. Scale bar, 200 μm and 50 μm. **C−E** Boxplots illustrating the fraction of PD1+ (**C**), TIM-3+ (**D**), and GZMB+ (**E**) CD8+ T cells in AM (yellow; Acral1, Acral2, Acral3-pre, Acral3-Post, Acral4) and CM (blue; Cutaneous1, Cutaneous2, Cutaneous1-lym), respectively. Box center lines, bounds of the box, and whiskers indicate medians, first and third quartiles, and minimum and maximum values within 1.5×IQR (interquartile range) of the box limits, respectively. Significance was determined using a two-sided, unpaired Wilcoxon rank-sum test relative to AM ($n = 25$ fields) for CM ($n = 15$ fields, P-value = 2.204e-07, 0.0001293, 5.966e-10, respectively). Source data are provided as a Source Data file. **F** Flow chart of drug treatment experiment at cellular level. **G, H** AM cells (**G**) and CM cells (**H**) were untreated, treated with anti-*PD1*, treated with anti-*TIM-3*, treated with anti-*PD1* and anti-*TIM-3* for 24 h and then labeled with Annexin V-FITC and PI-PE. The bar-plots represent the ratio of early apoptosis (Q3) and total apoptosis (Q2 + Q3) across for each treatment group.

Gene Set Enrichment Analysis (ssGSEA)[35] was used to calculate separate enrichment score of function module for each melanoma sample in TCGA[14]. The melanoma cohort was clustered subgroup using Non-negative Matrix Factorization (NMF) algorithm[36].

## Cell developmental trajectory

The cell lineage trajectory of CD8+ T was inferred by using Monocle2[37]. We excluded CD8-HOPX and CD8-EOMES cells according to their TCR identity due to their distinct TCRs and development processes relative to other CD8+ cells. Firstly, we used the "relative2abs" function in Monocle2 to convert TPM into normalized mRNA counts and created an object with parameter "expressionFamily = negbinomial.size" following the Monocle2 tutorial. We used the "differentialGeneTest" function to derive DEG from each cluster and genes with a *q*-value < 1e-5 were used to order the cells in pseudotime analysis. After the cell trajectories were constructed, differentially expressed genes along the pseudotime were detected using the "differentialGeneTest" function.

## Pathway analysis and functional annotation

We used Gene Ontology enrichment analysis and Single-sample Gene Set Enrichment Analysis (ssGSEA) for functional analysis. Gene signatures scores of samples were evaluated using R package GSVA. GO and KEGG analyses were performed by applying the "clusterProfiler" package.

## Definition of cell scores and signatures

We used the average expression (measured by log2 (CPM+ 1) of 5 resident markers (*RUNX3, NR4A1, CD69, CXCR6,* and *NR4A3*), 7 cytotoxicity associated genes (*PRF1, IFNG, GNLY, NKG7, GZMB, GZMA, CST7,* and *TNFSF10*), 5 exhausted markers (*CTLA4, HAVCR2 /TIM-3, LAG3, PDCD1/PD-1,* and *TIGIT*) and 6 costimulatory molecular genes (*ICOS, CD226, TNFRSF14, TNFRSF25, TNFRSF9,* and *CD28*) to define the resident, cytotoxic, exhausted, and costimulatory score for CD8+ T cells. We used non-negative matrix factorization algorithm implemented in R package NMF (v 0.23.0) to extract the characteristics of samples based on the identified gene module scores. The features of malignant cells were defined with the mean log2 (CPM+ 1) normalized expression of 5 modules signature genes.

## Survival analysis

Kaplan-Meier survival was used to analyze the prognosis of between groups by R package survival (v 3.1.12). We used the log-rank test to calculate differences of survival curves.

## Identification of differentially expressed genes

We considered CD8+ T cells from different states of Acral3 sample to identify the DEGs between before and after the PD-1 treatment. The "limma" package was used to identify DEGs with a *P* value < 0.05 and logFC > 0.5.

## Multiplex immunofluorescence staining

For fluorescent multiplex immunohistochemistry (mIHC) analysis, a five-color fluorescence kit based on tyramide signal amplification (TSA) were used following the manufacturer's protocol (abs50013, Absin Bioscience, China)[38]. In brief, tissue sections were incubated with primary antibodies as described in the above IHC protocol in two or three sequential cycles before application of corresponding secondary antibodies (PerkinElmer) and TSA solution for AlexaFluor488, AlexaFluor555, AlexaFluor594 and AlexaFluor647, respectively. The following primary antibodies were used: anti-CD8A (1:200, CST, catalog number: 85336, Clone numbers: D8A8Y), anti-CD4 (1:200, Abcam, catalog number:ab133616, Clone numbers: EPR6855), anti-CD20 (1:200, Abcam, catalog number: ab78237, Clone numbers: EP459Y), anti-FOXP3 (1:200, CST, catalog number: 98377, Clone numbers: D2W8E™), anti-TIM3 (1:200, Abcam, catalog number: ab241332, Clone numbers: EPR22241), anti-PD1 (1:200, CST, catalog number: 86163, Clone numbers: D4W2J), anti-GZMB (1:200, Abcam, catalog number: ab255598, Clone numbers: EPR22645-206), anti-CCR7(1:200, Abcam, catalog number: ab32527, Clone numbers: Y59). After the last TSA cycle, DAPI was counterstained at a dilution of 1:1000 for 10 min. Fluorescent images (300 ms exposure time) were obtained with an AxioImager.Z2 microscope (Carl Zeiss, Germany).

## Sorting CD8+ T cells and melanoma tumor cells

Fresh melanoma tissues were obtained from AM and CM patients. The tissues were minced, and trypsin was added, followed by shaking for 2 h and filtering through a 40 μm mesh to obtain a single-cell suspension. Human CD8+ immunomagnetic bead solution (Miltenyi Biotec) was added to single-cell suspension. The samples were incubated for 15 min at 4 °C and washed once with buffer, and the suspension cells were passed through the MS column in the magnetic field. Then, the column was removed from the magnet and 1 mL of wash buffer was added to the top of the column and the plunger (in the same package as the column) was immediately used to force the buffer through the column. The collected cells were used for the subsequent experiments, as CD8+ T cells.

Human tumor cell bead solution (Miltenyi Biotec) was added to single-cell suspension. The samples were incubated for 15 min at 4 °C and washed once with buffer, and the suspension cells were passed through the MS column in the magnetic field. The collected cells were used for the subsequent experiments, as melanoma tumor cells.

## Drug incubation of CD8+ T

We divided CD8+ T cells into four groups, including no drug added, 10 μg/mL *PD1* inhibitor added, 10 μg/mL *TIM-3* inhibitor added, 10 μg/mL *TIM3* inhibitor and 10 μg/mL PD1 inhibitor added, and incubated for 1 Hours.

## Co-culture of CD8+ T cells with melanoma tumor cells

To investigate whether CD8+ T cells can more effectively kill tumor cells after receiving immune checkpoint inhibitor treatment. Four groups of $10^5$ CD8+ T cells and tumor cells at a ratio of 5:1 were supplemented with 10% foetal bovine serum (FBS) (containing 60 mg/L penicillin and 100 mg/L streptomycin) and co-cultured at 37 °C in a 5% $CO_2$ incubator for 24 hours.The apoptosis of melanoma tumor cells was assessed by using an Apoptosis Kit (BD Biosciences).

## Detection of apoptosis by Flow cytometry analysis

An apoptosis assay (FITC Annexin V Apoptosis Detection Kit I, BD Biosciences) was used to detect apoptosis of melanoma tumor cells co-cultured with CD8+ T cells. The cells were washed twice with cold PBS

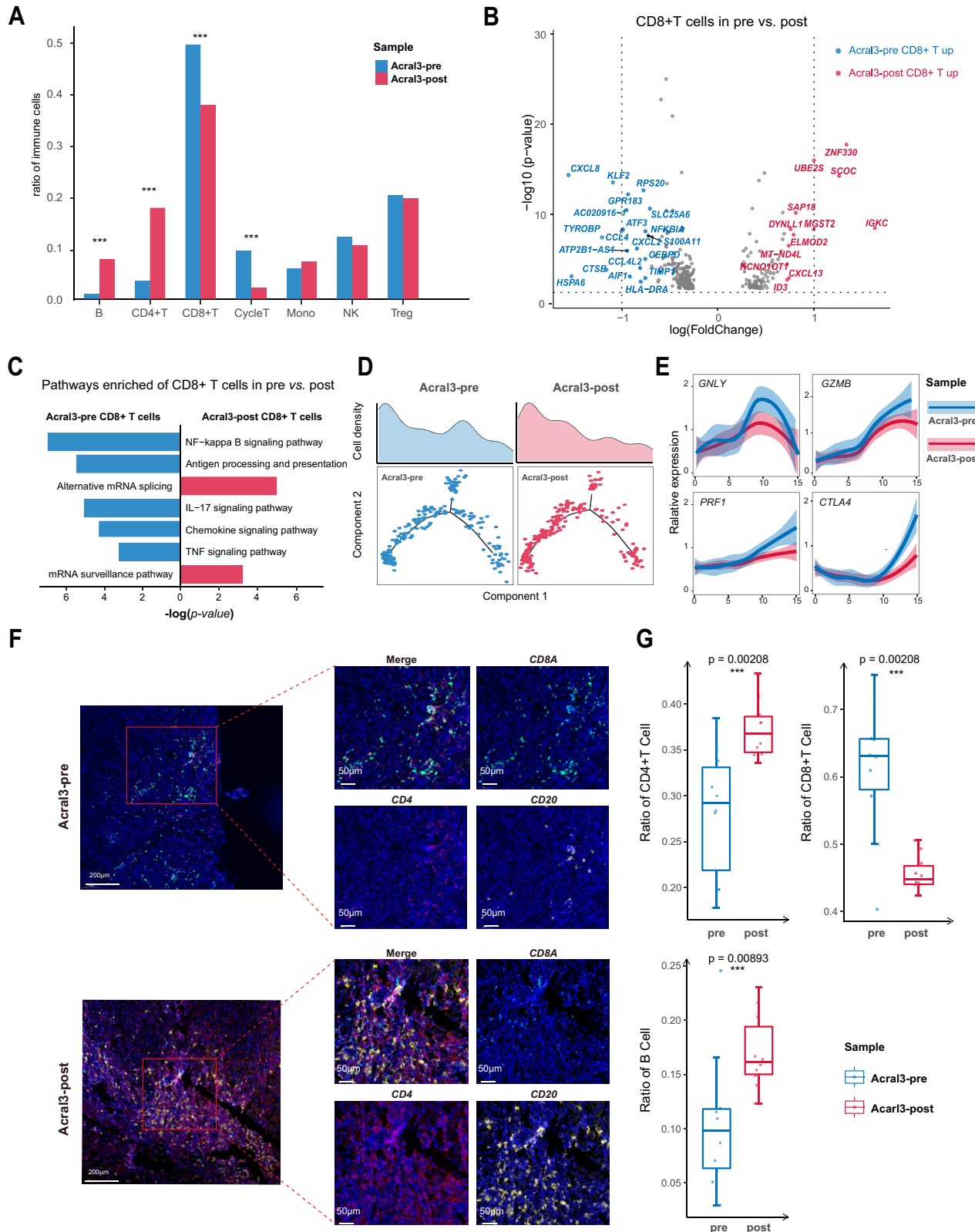

and were resuspended in 1× binding buffer at a concentration of $1 \times 10^6$ cells/ml. Then, 100 μL of the solution ($1 \times 10^5$ cells) was transferred to a 5-mL culture tube, and 5 μL of FITC Annexin V and 5 μL of PI were added. The cells were gently vortexed and incubated for 15 minutes at room temperature (25 °C) in the dark. Finally, 400 μL of 1× binding buffer was added to each tube. Analysis was performed by flow cytometry (Beckman Coulter).

## CNV analysis of single-cell

The large-scale chromosomal copy number variations of melanoma tumor cells were identified using inferCNV R routine (version 1.5.1). The chromosomal region in genetic profiles was inferred based on the average expression of single-cells. A random sample of 1000 tumor cells were taken from each sample, and stromal cells (1938 endothelial cells and 1624 fibrocytes) were taken into control group. Other

**Fig. 6 | Differences in pre-treatment and post-treatment samples from an immune-resistant patient with acral melanoma. A** Histogram showing the difference in ratio of immune cells subtypes between the pre-treatment and post-treatment samples. Significance was calculated using the two-side, unpaired Fisher's exact test (***P-value < 0.05, B cell P-value = 4.54e-09, CD4+ T cell P-value = 1.14e-14, CD8+ T P-value = 1.36e-05, Cycle T cell P-value = 1.36e-08, Monocyte and Macrophage P-value = 0.351, NK cell P-value = 0.380, Treg P-value = 0.898). Source data are provided as a Source Data file. **B** Volcano plot shows differentially expressed genes between Acral3-pre CD8+ T cells (blue dots) and Acral3-post CD8+ T cells (red dots). The names of the most significant genes are indicated in the plots. DEG were identified using MAST (only upregulated genes with p value < 0.001 and absolute logFC > 0.5) relative to Acral3-pre (n = 300 CD8+ T cells) for Acral3-post (n = 286 CD8+ T cells). Source data are provided as a Source Data file. **C** Two-sided bar graph showing the enriched activated pathways between CD8+ T cells of the Acral3-pre and Acral3-post. Source data are provided as a Source Data file. **D** 2D graph of the pseudotime-ordered CD8+ T cells, from Acral3-pre (left panel) and

Acral3-post (right panel) samples. The cell density distribution, by state, is shown at the top of the figure. Source data are provided as a Source Data file. **E** Dimensional plots showing the dynamic expression of cytokines genes and checkpoint genes during the T cell transitions along the pseudo-time. Error bands show local polynomial regression and the 95% confidence interval, respectively. **F** Representative images of Multiplex immunofluorescence staining in Formalin-fixed paraffin-embedded (FFPE) tissues, indicating CD8A+, CD4+, and CD20+ cells, in paired AM and CM samples. Scale bar, 200 μm and 50 μm. **G** Boxplots illustrating the fraction of CD8A+, CD4+, and CD20+ cells in AM (yellow; Acral1, Acral2, Acral3-pre, Acral3-Post, Acral4) and CM (blue; Cutaneous1, Cutaneous2, Cutaneous1-lym), respectively. Box center lines, bounds of the box, and whiskers indicate medians, first and third quartiles, and minimum and maximum values within 1.5×IQR (interquartile range) of the box limits, respectively. Significance was determined using a two-sided, unpaired Wilcoxon rank-sum test relative to Acral3-pre (n = 10 fields) for Acral3-post (n = 10 fields, P-value = 0.00208, 0.00208, 0.00893, respectively. Source data are provided as a Source Data file.

---

parameters were set as default. The 44 differentially expressed genes were screened by comparing the expression of all genes on chromosome 4 in the samples before and after immunotherapy treatment. The comparisons of patients with different immunotherapy RECIST outcome were performed using the gene expression variation between pretherapy and post-treatment.

## Statistics analysis

Cell distribution comparisons between AM and CM were performed using Wilcoxon rank-sum test or Fisher's exact test. All statistical analyses and presentation were performed using R. Statistical tests used in figures were shown in figure legends and statistical significance was set at $p < 0.05$ or adjusted $p < 0.1$. Two-sided test was used if not specified.

## Reporting summary

Further information on research design is available in the Nature Portfolio Reporting Summary linked to this article.

## Data availability

The raw DNA sequencing, TCR sequencing and single-cell RNA sequencing data generated in this study have been deposited in the Sequence Read Archive (SRA) database under accession code PRJNA862451. The raw bulk RNA sequencing data generated in this study have been deposited in the Gene Expression Omnibus (GEO) database under accession code GSE215121. The processed single-cell/bulk RNA data are available at the GEO database under accession code GSE215121. The remaining data generated in this study are provided in the Supplementary Information or Source Data file. Source data are provided with this paper. The publicly data used in this study are available in the GEO database under accession code GSE72056, GSE189889, and GSE120575 Source data are provided with this paper.

## Code availability

The source code of Miscell is available at https://github.com/lixiangchun/Miscell.

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

## Acknowledgements

We thank professor Wei Zhang of Wake Forest Baptist Comprehensive Cancer Center Wenyi Mi and Jun Li of Tianjin Medical University, Cihui Yan, Jinpu Yu, Xiubao Ren, and Wenwen Yu of Tianjin Medical University Cancer Hospital for their valuable suggestions and experiment guidance of this article. This work was supported by the Science & Technology Development Fund of Tianjin Education Commission for Higher Education (2021KJ199;T.L.).

## Author contributions

C.Z., H.S., Yichen.Y., M.Y., D.W., and Y.L. carried out the analysis. T.Y., T.L., H.L., X.L., Junqiang.W., Z.L., Jin.W., J.L., and L.X. performed the experimental studies. R.X., S.T., J.Z., Yun.Y., G.Z., K.C., X.L., and J.Y. supervised the work.

## Competing interests

The authors declare no competing interests.
