## [Peer Review File · Nature Communications]

A single-cell analysis reveals tumor heterogeneity and immune environment of acral melanomaReviewers' Comments:

Reviewer #1:

Remarks to the Author:

A single cell atlas reveals tumor heterogeneity and immune environment of acral melanoma

This paper presents single cell seq (SCS) data of 5 acral and 2 primary cutaneous melanomas with a matched LN met. This is an exceptional resource because these are results from a North Asian population, and there are no previous SCS of primary cutaneous or acral from this population. Therefore, the data are intriguing to the melanoma community.

However, there are significant and fundamental weaknesses. Scientifically, these results have to be contrasted to Northern European data, such as the Tirosh resource of SCS metastasis study, or at the very least large bulk RNAseq for primaries, mets, therapy responses. The study as presented is a mere descriptive download of data. Moreover there are incorrect premises, no context of the field and no rationales or hypotheses put forward.

A second major weakness is the writing which is grammatically incorrect and at points incomprehensible.

1. in the abstract: the authors state that acral mm occurs in extremities of darker- skinned people. This is incorrect. Acral mm occurs in equal numbers across all ethnicities, and only stand out in the non-European community as there is a dearth of cutaneous mm in these groups. This is a key point which means their cutaneous data must be contrasted to fair skinned population data- the true population difference. Moreover, they state incidence numbers of acral mm as proportions, which is not an incidence, and erroneously states acral is more prevalent in Asia.

2. They identify 50 clusters of immune and non immune cells. How do the clusters compare to other studies. Can they use their immune markers at least to study within the TCGA the levels of expression, normalised to stroma, to infer similarities? The TCGA has numerous acral and cutaneous primaries, as well as vast metastases. They must provide a comprehensive look within the numerous studies presenting responses to IT as well for this paper to be valuable and to raise true significant contextual findings.

3. The immune profile of acral mm is unclear from the figures, data and writing. How exactly do they conclude that the cd8 profile is exhausted if they use time trajectories. Are these acral mm simply not more advanced, a common endpoint for all samples. How do these data compare to Tcell profile of responses. What are the TCR characteristics of these samples compared to each other if they are indeed more exhausted. How do these profiles correlate to cell cycle gene expression, considering the novel links published in at least 2 nature comms studies between cell cycle – immunity – immunotherapy? Also, in the discussion they speak about T cell exhaustion in the cutaneous mm samples. How do they differ, what are the possible biological mechanisms driving these differences?

Minor – 2 examples of lack of clarity from the text – this is throughout the paper.

1. It is unclear which six patients with acral mm did not respond to KIT inhibition, presented in the introduction.

2. They state acral melanoma has most mutations with a UV signature? The text is riddled with mistakes that are pretty fundamental.

Reviewer #2:

Remarks to the Author:

In this manuscript, the authors have performed single-cell RNAseq analysis of a small cohort of acral

(n= 5 samples from 4 patients) and cutaneous (n=3 tissues from 2 patients) melanomas with the goal of characterizing the differences in the tumor cells and the immune environment between acral and cutaneous melanomas. The authors have found acral melanoma specific gene signatures in the tumor cells as well as differences in the immune infiltration of the tumor microenvironment. The acral immune environment is shown to be "cold" compared to cutaneous melanoma, including differences in Tregs, cytotoxic T cells, and exhausted T cells (as well as differences in the checkpoints they express). Overall, this is a valuable study which could contribute to better understanding of the biology of acral melanoma and provide rationale for more efficacious, acral-specific therapies. However, there are several key limitations which should be addressed in order for this study to be suitable for publication. Summary of key weaknesses include the incredibly limited sample number and lack of robust validation and functional studies.

1. Due to the limited number of specimens analyzed here (as well as a single scRNAseq experiment used for the majority of the paper), the authors need to validate some key findings in additional specimens including:

- a. Please provide staining of clinical specimens that show Tregs and naïve/exhausted T cells to be more abundant in acral melanoma specimens. Please provide staining to show more cytotoxic T cells present in cutaneous melanoma specimens.
- b. The majority of the results in Figure 5 come from a single pair of pre-and post-treatment specimens. The authors need to validate the differential presence of more CD8+ T, Cytotoxic T cells, CD4+ T and B cells in a larger cohort of patient specimens.

2. Please provide the mutational status of each patient, this is critical when comparing melanoma specimens. The biology of melanoma is heavily dependent on the driver mutation and is therefore essential in the understanding of these comparisons.

3. The authors aim to show that the transcriptional signatures of melanoma cells are associated with patient prognosis/melanoma subtype but there are a few weaknesses which should be addressed:

- a. The authors split up the prognosis based on whether the patients are deceased or alive. This may be entirely irrelevant relative to their prognosis or clinical outcomes. A better measure would be survival time from diagnosis. For example, in a comparison of two patients, patient 1 may be deceased and patient 2 may be alive at a given time but the overall survival time of patient 1 may be much longer than patient 2 if they were diagnosed much earlier (time-wise). Looking at the supplemental Table 1, it is clear that two of the 4 patients who are dead had a relatively long survival time (two of the longest in the cohort!). Therefore, it is not appropriate to use dead/alive as the two comparison groups.
- b. Please perform statistical testing to show the association of transcriptional signatures with patients prognosis/survival time (related to Figure 2G). By eye, signature 3 doesn't seem to be enriched in any of the samples from either of the two groups.
- c. The p-value on the log-rank test of the survival curves in Figure 2H is 0.2, meaning there is no difference in survival probability between the two clusters of signatures.
- d. Which samples are which on Figure 2G? Which of them are from patients that are deceased? What are their individual survival times? This information should be labelled on the figure for easy/quick reference. It's unclear from the data whether the gene signatures simply separate the acral and cutaneous melanoma (and therefore account for the differences in survival). It's possible the signatures themselves are not related to survival but rather related to melanoma subtype.
- e. Please perform a statistical test to the association of transcriptional signatures with patients melanoma subtype (related to Figure 2K).

4. Please provide a supplementary figure for the data mentioned: "In addition, CD8-MT1E, which is a new CD8+ T cell 180 subgroup, was characterized by high expression of MT1E and MT2A and enriched in 181 two deceased acral melanoma patients." It's not immediately clear if any of the included figures demonstrate this explicitly.

5. Could the authors provide any statistical test to support the conclusion that exhausted CD8+ T cells of acral melanoma patients were characterized by high expression of PDCD1 (PD-1) and HAVCR2 (TIM-3) relative to cutaneous patients (Figure 4G)?

6. Please provide statistics for Figure 5A.

7. Supplementary Figure 1B also shows that there appears to be a general decrease in checkpoint genes after therapy, could the authors comment on this?

8. The authors claim "The differential expression levels of these 44 genes from clinical trial (NCT01621490) dataset were higher in patients resistant to immunotherapy (PD, evaluated by Recist1.1) than those who are responsive¹⁴ (CR and PR, Figure 5H)" but Figure 5H does not show any clear differential expression of these genes. Authors need to show statistical analysis of this 44-gene signature between response groups.

9. Please provide a table of the 44 genes found to be amplified on chromosome 4 post treatment. Also provide the number of patients analyzed from study NCT01621490 for each category (PD, PR and CR, etc).

10. This work completely lacks any functional studies, such as animal models to show that immune therapy of acral melanoma leads to less CD8+ T cells, or that targeting PD1 plus TIM-3 is more appropriate for acral melanoma.

Minor:

Please include sequencing metrics for each sample in the single cell RNAseq experiment.

Please include a key for the color rectangles in Figure 3B.

Please provide the demographics summary for melanoma patients analyzed for Figure 2L, what is the average age, gender, etc? Do you typically see early stage patients or late-stage patients? This would be really helpful for comparison to other demographics. For example, in western countries the acral patients tend to have worse outcomes than cutaneous.

There are a lot of grammar issues throughout the manuscript (especially in Introduction section) that need to be fixed, too many to list individually. For example: "The incidence of acral melanoma is approximately 50% in the Asian population and less than 5% in the European and American populations." Makes it sound like 50% of all Asian people get acral melanoma as opposed to the fact that 50% of all melanomas in the Asian population are acral melanomas.

Please re-label Figure 1 panels in chronological appearance in manuscript (ie panel 1D is mentioned before panel 1C and should switch label).

The authors should comment a bit further on improved survival observed in acral melanoma patients vs cutaneous at their institution in the context of worse therapy responses observed in acral patients. In their introduction they point out that acral melanoma therapy responses are much worse compared to those of cutaneous and go on to describe a more immune-cold environment of acral melanoma compared to cutaneous. What could account for the improved survival in acral patients?

Reviewer #3:

Remarks to the Author:

Zhang et al present a single-cell analysis of acral melanomas and cutaneous melanoma from 4 and 2 patients, respectively, including one patient with acral melanoma with pre/post sample on PD1 therapy.

They propose the presence of melanoma subgroups with distinct representation in acral vs. cutaneous melanoma; different cell type proportions; increased abundance of Tregs in acral; and varying cell states among CD8 + T cells; they argue that T cells from acral are more likely to be exhausted. In a one-patient pre/post comparison they find pathways and inferred genomic alterations which they link to response outcomes.

Treatment of acral melanoma is extremely challenging given the low response rates to immunotherapies and absence of druggable oncogenic drivers, as such, this is an important topic. However the study has serious limitations.

First, the sample sizes are simply too small. I recognize that collection of such samples is challenging, but no meaningful comparisons can be drawn from comparing 4 vs. 2 patients, wherein repeated sampling of the same patient (two in the acral group and one in the cutaneous group) is not even accounted for. The inability to apply proper statistics is pervasive in the entire manuscript and results in a lot of descriptions, such as "higher abundance of cell type X" without any statistical evaluation of these statements. When statistical evaluations are applied (e.g. abundance of Tregs, Fig. 3d) there is no correction for multiple hypothesis testing, which would be dismal in any of these cases due to low sample size (in fact, only $n=2$ in the cutaneous group)

Second, the clustering as shown clearly demonstrates that the data suffers from serious batch effects that are not accounted for. Furthermore, the boundaries for clusters (e.g. Fig 2a) seem arbitrary and suffer from serious batch effects (e.g. cluster 1 made up completely by one patient). Further evidence for batch effects are analyses presented in figure 4D, clearly showing that T cell populations are extremely variable for the exhausted score, indicating that this effect is driven by individual samples.

Third, the pre/post comparison using genomic inference is a bit overinterpreted. Only because cells with chr. 4 were not detected in the pre-treatment specimen, does not prove that they were not present. the tool they use to narrow down on gene expression (inferCNV) is furthermore not designed to define small gene segment windows, yet they use the "differential expression" of genes on Chr 4 in post vs pre to make further inferences.

Overall, my strong recommendation is that the authors consider these weaknesses, increase the sample sizes when possible, or otherwise stick to a purely descriptive approach to interpreting this data.

Reviewer #1 (Comments to the Author (Required)):

A single cell atlas reveals tumor heterogeneity and immune environment of AM.

This paper presents single cell seq (SCS) data of 5 acral and 2 primary CMs with a matched LN met. This is an exceptional resource because these are results from a North Asian population, and there are no previous SCS of primary cutaneous or acral from this population. Therefore, the data are intriguing to the melanoma community. However, there are significant and fundamental weaknesses. Scientifically, these results have to be contrasted to Northern European data, such as the Tirosh resource of SCS metastasis study, or at the very least large bulk RNAseq for primaries, mets, therapy responses. The study as presented is a mere descriptive download of data. Moreover, there are incorrect premises, no context of the field and no rationales or hypotheses put forward.

A second major weakness is the writing which is grammatically incorrect and at points incomprehensible.

Response:

Thanks for your comments. Per the suggestion, we collected transcriptome data from 2 single-cell studies on melanoma conducted by Tirosh and colleagues¹ and Sade-Feldman and colleagues², bulk tumor transcriptome data from TCGA SKCM³ and NCT01621490 clinical trial⁴. The NCT01621490 clinical trial dataset includes samples of pre- and post-immunotherapy treatment. In addition, we added RNA-seq data of 57 melanoma samples (including 15 CMs and 42 AMs) collected from Tianjin Medical University Cancer Hospital. We compared our data with these collected data and revised the main text accordingly. We revised and polished our manuscript. Now we have 6 figures and 6 supplemental figures.

1 In the abstract: the authors state that acral mm occurs in extremities of darker-skinned people. This is incorrect. Acral mm occurs in equal numbers across all ethnicities, and only stand out in the non-European community as there is a dearth of cutaneous mm in these groups. This is a key point which means their cutaneous data must be contrasted to fair skinned population data- the true population difference. Moreover, they state incidence numbers of acral mm as proportions, which is not an incidence, and erroneously states acral is more prevalent in Asia.?

Response:

Thanks for your comment. We have revised the abstract and main text accordingly. We compared clinical data of 251 CM patients from Tianjin Medical University Cancer Hospital (TMCH-CM) to the TCGA-SKCM cohort. There were significant

differences in TMCH-CM cohort versus TCGA-SKCM cohort with respect to age, sex, TNM stage and overall survival (**Response Figure 1A-B**).

Response Figure 1. Comparison between TMCH-CM and TCGA-SKCM in terms of Age, Sex, TNM stage and overall survival. (A) Statistical table of clinical information for TMCH and TCGA datasets, including age, sex, clinical stage, and statistical differences. (B) Kaplan-Meier analysis showing the overall survival rate of TMCH-CM and TCGA patients, characterized by TMCH-CM (dark blue) and TCGA (red). The numbers of patients and the risk classification are indicated in the figure. Significance was calculated using the log-rank test.

2 They identify 50 clusters of immune and non immune cells. How do the clusters compare to other studies. Can they use their immune markers at least to study within the TCGA the levels of expression, normalised to stroma, to infer similarities? The TCGA has numerous acral and cutaneous primaries, as well as vast metastases. They must provide a comprehensive look within the numerous studies presenting responses to IT as well for this paper to be valuable and to raise true significant contextual findings.

Response:

Thanks for your comments. We observed that the markers identified in our study was able to distinguish different cell types as defined by Tirosh and colleagues in their study¹ (**Figure 1C**), and we added **Figure S2** in the main text. In the TCGA-SKCM cohort, detailed information of AM and CM is not available. The sites of primary tumor given by TCGA-SKCM are only categorized into trunk and extremity.

We analyzed the scRNA-seq data of 31 samples that were not responsive to immunotherapy treatment from Sade-Feldman and colleagues. We observed that the ratio of CD4+ T cells are significantly higher in post-treatment samples than pre-treatment samples. The ratio of CD8+ T cells are less represented in post-treatment samples than pre-treatment samples. This observation is consistent

with our finding obtained from one patient that was not responsive to immunotherapy treatment (**Figure S6C**). Ratio of these immune cells in response group is provided in **Figure S6D**.

In the tissue sequencing samples from NCT01621490 clinical trial⁴, for 18 patients that were not responsive to IT treatment (PD), we also observed that higher ratio of CD4+ T cells and lower ratio of CD8+ T cells in post-treatment samples as compared with pre-treatment samples (**Figure S6E**). Ratio of these immune cells in response group is provided in **Figure S6F-H**.

3. The immune profile of acral mm is unclear from the figures, data and writing. How exactly do they conclude that the cd8 profile is exhausted if they use time trajectories. Are these acral mm simply not more advanced, a common endpoint for all samples. How do these data compare to T cell profile of responses. What are the TCR characteristics of these samples compared to each other if they are indeed more exhausted. How do these profiles correlate to cell cycle gene expression, considering the novel links published in at least 2 nature comms studies between cell cycle – immunity – immunotherapy? Also, in the discussion they speak about T cell exhaustion in the cutaneous mm samples. How do they differ, what are the possible biological mechanisms driving these differences?

Response:

Thank you very much for your question. The conclusion of exhausted CD8 profile is not based on time trajectories. We defined the exhausted CD8 profile according to the expressions of immune checkpoint genes adopted by other study⁵ such as *CTLA4*, *PDI*, *TIM-3*, *LAG3* and *TIGIT*. These five immune checkpoint genes are indeed high expressed as illustrated by Figure 4B and E. We are not able to perform single-cell TCR-seq due to the unavailability of the fresh tissues of these 8 samples after scRNA-seq. We performed TCR-seq of FFPE tissue for these 8 samples, however, two of them failed including the post-treatment sample Acral3-post. The TCR characteristics of these FFPE samples are that the number of clonotypes presented a lower level in the 4 AM samples compared to 2 CM samples (**Figure S4A-B**). Clonal diversity in AM is lower as compared with CM (**Figure S4D-F**). We calculated the GSEA score for T cell signatures defined in **Figure 3C** and cell cycle circuit on the NCT01621490 clinical trial dataset and analyzed the correlation between these T cell signatures and cell cycle. The result showed that the transcriptional factors (TF) signature is negatively correlated with cell cycle (**Response Figure 2A**). No significant difference was found for cell cycle GSEA score between response and non-response patients (**Response Figure 2B**).

Response Figure 2. The associations of cell cycle signature with different T cell signatures and immunotherapy response. (A) The correlations of cell cycle signature score and 6 T cell signature scores. (B) Boxplot of the Signature3 scores between response and non-response patients.

In general, CD8⁺ T cells of AM were more exhausted than CM in terms of exhausted

scores of CD8+ T cells (**Figure 4D**). The added multiplex immunofluorescence staining results indicated that CD8+ T cells of AM had higher expression of *TIM-3* and *PDI* (**Figure 5A-E**) than CM. CM had higher expression of *CTLA4*, *TIGIT* and *LAG3* whereas AM had higher expression of *TIM-3* and *PDI*. This indicates different mechanisms underlying the exhausted state of CM versus AM. In addition, the environmental factor also contributed to the different exhausted state of AM and CM in that UV exposure is dominated in the tumorigenesis of CM but not AM.

Minor – 2 examples of lack of clarity from the text – this is throughout the paper.

1. It is unclear which six patients with acral melanoma did not respond to KIT inhibition, presented in the introduction.

Response:

Thank you for your careful review of the article. These 6 AM patients are reported in the other study. We revised the main text as follows to avoid the ambiguity: “a phase II clinical trial showed that Imatinib was ineffective for AM with KIT mutations”

2. They state AM has most mutations with a UV signature? The text is riddled with mistakes that are pretty fundamental.

Response:

Thanks for your careful review. We have revised the introduction.

Reviewer #2 (Comments to the Author (Required)):

In this manuscript, the authors have performed single-cell RNAseq analysis of a small cohort of acral (n= 5 samples from 4 patients) and cutaneous (n=3 tissues from 2 patients) melanomas with the goal of characterizing the differences in the tumor cells and the immune environment between acral and cutaneous melanomas. The authors have found acral melanoma specific gene signatures in the tumors cells as well as differences in the immune infiltration of the tumor microenvironment. The acral immune environment is shown to be “cold” compared to cutaneous melanoma, including differences in Tregs, cytotoxic T cells, and exhausted T cells (as well as differences in the checkpoints they express). Overall, this is a valuable study which could contribute to better understanding of the biology of acral melanoma and provide rationale for more efficacious, acral-specific therapies. However, there are several key limitations which should be addressed in order for this study to be suitable for publication. Summary of key weaknesses include the incredibly limited sample number and lack of robust validation and functional studies.

Response:

Thank you very much for your positive comments. Because of the low incidence rate of acral melanoma, we only have collected 8 samples for 3 years, and conducted in-depth analysis. During the following six months, we did treat some acral melanoma patients, but we could not add single cell sequencing data because of lack high quality fresh tissues. In order to solve the research limitations, we added transcriptome sequencing of 57 melanoma samples (15 Cutaneous melanoma and 42 Acral melanoma), immunofluorescence staining, drug treatment at the cellular level and validation analysis on external datasets during this period to further verify the analysis results of single-cell transcriptome. We revised and polished our manuscript. Now we have 6 figures and 6 supplemental figures.

1. Due to the limited number of specimens analyzed here (as well as a single scRNAseq experiment used for the majority of the paper), the authors need to validate some key findings in additional specimens including:

a. Please provide staining of clinical specimens that show Tregs and naïve/exhausted T cells to be more abundant in acral melanoma specimens. Please provide staining to show more cytotoxic T cells present in cutaneous melanoma specimens.

Response:

Thanks for your comments. We performed multiplex immunofluorescence staining on the tissue sections of 8 samples mentioned in this paper, and confirmed the results revealed by our single-cell data. As shown in **Figure 3E**, the proportion of *FOXP3*+ Treg cells in the infiltrated immune cells in the tumor tissue of acral melanoma was

significantly higher than that of cutaneous melanoma. And we selected 5 fields of view for each sample for statistics, and the results showed that there were more Treg cells in the immune infiltration of acral melanoma (**Figure 3F**). At the same time, the proportion of exhausted CD8+ T cells (markers as *TIM-3* and *PD1*) was also significantly higher (**Figure 5A-D**). On the contrary, cytotoxic CD8+ T cells with high expression of *GZMB* are more enriched in the tumor environment of cutaneous melanoma (**Figure 5A, 5B and 5E**). The staining of *CCR7* was not available due to non-specific staining output.

b. The majority of the results in Figure 5 come from a single pair of pre-and post-treatment specimens. The authors need to validate the differential presence of more CD8+ T, Cycle T cells, CD4+ T and B cells in a larger cohort of patient specimens.

Response:

Thanks for your comments. We have verified the differential presence of immune cells in two independent datasets and immunostaining.

1. We analyzed the scRNA-seq data of 31 samples that were not responsive to immunotherapy treatment from Sade-Feldman and colleagues². We observed that the ratio of CD4+ T cells are significantly higher in post-treatment samples than pre-treatment samples. The ratio of CD8+ T cells are less represented in post-treatment samples than pre-treatment samples. This observation is consistent with our finding obtained from one patient that was not responsive to immunotherapy treatment (**Figure S6C**). Ratio of these immune cells in response group is provided in **Figure S6D**.

In the tissue sequencing samples from NCT01621490 clinical trial⁴, for 18 patients that were not responsive to IT treatment (PD), we also observed that higher ratio of CD4+ T cells and lower ratio of CD8+ T cells in post-treatment samples as compared with pre-treatment samples (**Figure S6E**). Ratio of these immune cells in response group is provided in **Figure S6F-H**. These also confirm our results to a certain extent.

It is worth noting that some of the changes in the proportion of immune cells pre- and post- immunotherapy presented by these two sets of data only show the same trend of changes, and some are not similar, such as the proportion of B cells. It may be because the patients who received immunotherapy in these two datasets were CM patients, and the patients who received immunotherapy and developed resistance in our paper were AM patients.

2. Our immunostaining results also confirmed this view again. As shown in the **Figure 6F-G**, there was a significant decrease in CD8+ T cell and a significant

increase in CD4+ T cells and B cells in the immune infiltration of AM patient who was not responsive to immunotherapy treatment.

In addition, considering that using a cell cycle-related marker to label cycle T, it will interfere with the expression of this marker in tumor cells, resulting in inaccurate results. Also due to the limited number of markers for multiplex immunofluorescence staining, we did not implement the ratio change of cycle T cell for verification. In the future we will find other methods to validate this issue.

2. Please provide the mutational status of each patient, this is critical when comparing melanoma specimens. The biology of melanoma is heavily dependent on the driver mutation and is therefore essential in the understanding of these comparisons.

Response:

Thanks for your comment. We performed whole-exome sequencing for these 8 samples. We presented the mutation landscape of canonical driver genes in the **Figure 1A**.

3. The authors aim to show that the transcriptional signatures of melanoma cells are associated with patient prognosis/melanoma subtype but there are a few weaknesses which should be addressed:

a. The authors split up the prognosis based on whether the patients are deceased or alive. This may be entirely irrelevant relative to their prognosis or clinical outcomes. A better measure would be survival time from diagnosis. For example, in a comparison of two patients, patient 1 may be deceased and patient 2 may be alive at a given time but the overall survival time of patient 1 may be much longer than patient 2 if they were diagnosed much earlier (time-wise). Looking at the supplemental Table 1, it is clear that two of the 4 patients who are dead had a relatively long survival time (two of the longest in the cohort!). Therefore, it is not appropriate to use dead/alive as the two comparison groups.

Response:

Thanks for your comment. We revised the description in the main text on Page 6.

b. Please perform statistical testing to show the association of transcriptional signatures with patients prognosis/survival time (related to Figure 2G). By eye, signature 3 doesn't seem to be enriched in any of the samples from either of the two groups.

Response:

Thanks for your comment. We added statistical testing and updated the **Figure 2G**. We revised the main text accordingly.

c. The p-value on the log-rank test of the survival curves in Figure 2H is 0.2, meaning there is no difference in survival probability between the two clusters of signatures.

Response:

Thanks for your comments. This is probably due to the limited sample size. The significant difference between C1 and C2 was verified on the added internal bulk RNA-seq of 57 samples (See updated **Figure 2I** and **J**).

d. Which samples are which on Figure 2G? Which of them are from patients that are deceased? What are their individual survival times? This information should be labelled on the figure for easy/quick reference. It's unclear from the data whether the gene signatures simply separate the acral and cutaneous melanoma (and therefore account for the differences in survival). It's possible the signatures themselves are not related to survival but rather related to melanoma subtype.

Response:

Thank you for your comments. Firstly, we have added the sample information in the revised **Figure 2G**. AM and CM are admixed in both C1 and C2.

e. Please perform a statistical test to the association of transcriptional signatures with patients melanoma subtype (related to Figure 2K).

Response:

Thanks for your comments. We have added the statistical test and revised the Figure accordingly. The original **Figure 2K** was moved to **Figure S3C-D** in the revised main text.

4. Please provide a supplementary figure for the data mentioned: "In addition, CD8-MT1E, which is a new CD8+ T cell 180 subgroup, was characterized by high expression of MT1E and MT2A and enriched in 181 two deceased acral melanoma patients." It's not immediately clear if any of the included figures demonstrate this explicitly.

Response:

Thanks a lot. We have provided **Figure S3L** in the revised main text to support the above statement accordingly.

5. Could the authors provide any statistical test to support the conclusion that exhausted CD8+ T cells of acral melanoma patients were characterized by high expression of PDCD1 (PD-1) and HAVCR2 (TIM-3) relative to cutaneous patients (Figure 4G)?

Response:

Thanks for your comment. The results are consistent with those explained in our paper. We added the statistical test for *PDI* and *TIM-3* and revised the main text as: “In the phase 3, *PDI* and *TIM-3* were highly expressed in AM (*PDI*, AM versus CM, $p<0.01$; *TIM-3*, AM versus CM, $p<0.01$)”. We also perform staining (**Figure 5A-D**).

6. Please provide statistics for Figure 5A.

Response:

Thanks for your comment. We have added statistics to **Figure 6A** accordingly.

7. Supplementary Figure 1B also shows that there appears to be a general decrease in checkpoint genes after therapy, could the authors comment on this?

Response:

Thanks for your comments. The proportion of CD8+ T cells in sample Acral3 is decreased after immunotherapy treatment. Immune checkpoint genes such as *CTLA4* have the effect of inhibiting lymphocyte proliferation. The decrease in the expression level of *CTLA4* may be a negative feedback regulation in response to the decrease of CD8+ T cells. Therefore, decreased CD8+ T cells might lead to decrease in checkpoint genes.

8. The authors claim “The differential expression levels of these 44 genes from clinical trial (NCT01621490) dataset were higher in patients resistant to immunotherapy (PD, evaluated by Recist1.1) than those who are responsive (CR and PR, Figure 5H)” but Figure 5H does not show any clear differential expression of these genes. Authors need to show statistical analysis of this 44-gene signature between response groups.

Response:

Thank you very much for your comments, we have completed the differential analysis of the expression changes of these 44 genes between the CR, PR group and PD group and added the statistical results to the (**Figure S5E**). Of these, 12/44 gene had significant differences. We compared the signature scores of 44 gene sets of patients in CR, PR group and PD group. It was found that the signature score of patients in PD

group was significantly higher than that in CR and PR group (**Response Figure 3**).

Response Figure 3. The associations of signature score of 44 genes with different immunotherapy response. Boxplot of the signature scores of 44 genes between CR+PR group and PD group.

9. Please provide a table of the 44 genes found to be amplified on chromosome 4 post treatment. Also provide the number of patients analyzed from study NCT01621490 for each category (PD, PR and CR, etc).

Response:

Thanks for your comment, we have provided a table of 44 genes as **Table S3**. The number of patients for study NCT01621490 are added to **Figure S5D**.

10. This work completely lacks any functional studies, such as animal models to show that immune therapy of acral melanoma leads to less CD8+ T cells, or that targeting PD1 plus TIM-3 is more appropriate for acral melanoma.

Response:

Thanks for your comments. Due to the lack of acral melanoma cell line, it is impossible to plant mouse tumor through cell line. If PDX model is used for planting, human tumor planting needs immune deficient mice. But drugs we used are aimed at immune cell surface receptors. It is also difficult to establish mouse derived acral

melanoma cell line in a short time. So we regret to confess that we could not get good animal models to validate at a period of six months required by the Journal.

However, we performed some experiments at cellular level to validate the results. We chose to extract tumor cells and CD8 + T cells from the fresh tumor tissues of AM and CM patients and co-culture them to verify this view (**Figure S1, Figure 5F-H**). We isolated tumor cells and CD8+ T cells from fresh tumor tissues of AM and CM patients, respectively. CD8+ T cells were divided into 4 groups and co-raised with immune checkpoint inhibitor drugs (including anti-*PD1*, anti-*TIM3*, anti-*PD1*+anti-*TIM3* and Control groups) for 1 hour, and then co-cultured with tumor cells for 24 hours and detected the apoptotic ratio of tumor cells (**Figure 7F**). The results showed that compared with the control group, the apoptosis ratio of AM patient's tumor cells were significantly increased in the anti-*PD1* and anti-*TIM3* groups, and the combination group was the most obvious (**Figure 5G**). The apoptosis ratio of CM patient's tumor cells was not obvious in the anti-*TIM-3* group, and the combination group was similar to that in the anti-*PD1* group (**Figure 5H**).

Minor:

1.Please include sequencing metrics for each sample in the single cell RNAseq experiment.

Response:

Thanks for your comment, we have included the sequencing information of the 8 single-cell samples in the study as **Table S5**.

2.Please include a key for the color rectangles in Figure 3B.

Response:

Thanks for your comment, we have included indicator bars for various types of immune cells in **Figure 3B**.

3.Please provide the demographics summary for melanoma patients analyzed for Figure 2L, what is the average age, gender, etc? Do you typically see early stage patients or late-stage patients? This would be really helpful for comparison to other demographics. For example, in western countries the acral patients tend to have worse outcomes than cutaneous.

Response:

Thank you very much for your comments, we have added the clinical information of 602 patients to **Table S4**. And the clinical data were analyzed for survival differences in 4 stages, and the results showed that the OS of acral melanoma was better than that

of cutaneous melanoma in all 4 stages (**Figure S3G-K**).

4. There are a lot of grammar issues throughout the manuscript (especially in Introduction section) that need to be fixed, too many to list individually. For example: “The incidence of acral melanoma is approximately 50% in the Asian population and less than 5% in the European and American populations.” Makes it sound like 50% of all Asian people get acral melanoma as opposed to the fact that 50% of all melanomas in the Asian population are acral melanomas.

Response:

Thanks for your suggestion, we have revised the description of the article and corrected some grammatical errors. We have highlighted the specific changes.

5. Please re-label Figure 1 panels in chronological appearance in manuscript (ie panel 1D is mentioned before panel 1C and should switch label).

Response:

Thanks for your suggestion, we have adjusted the order of the figures as requested.

6. The authors should comment a bit further on improved survival observed in acral melanoma patients vs cutaneous at their institution in the context of worse therapy responses observed in acral patients. In their introduction they point out that acral melanoma therapy responses are much worse compared to those of cutaneous and go on to describe a more immune-cold environment of acral melanoma compared to cutaneous. What could account for the improved survival in acral patients?

Response:

Thanks for your opinions and questions. In fact, this is also our concern. Our single cell and clinical data do show that the overall survival of AM is better than CM, but their immune environment is relatively poor. Firstly, all these 602 patients did not receive immunotherapy treatment. The difference in survival may come from the difference of tumor cells themselves, that is, as we analyzed in this paper, most tumor cells in AM are composed of signatures related to good prognosis (differentiation is relatively more mature), while CM is on the contrary. This survival difference stems from the nature of tumor cells themselves. However, the special immune environment (such as Treg, exhausted CD8+ T) of acral melanoma is the reason for the poor effect of immunotherapy. This does not seem to conflict, and in the studies of other institutions, the poor overall survival of AM patients may be the result of poor immunotherapy response, because CM patients benefits more from immunotherapy, which improves the current overall survival rate. We think this is not contradictory. However, it is still a controversy question and need further evidence and

investigation.

Reviewer #3 (Comments to the Author (Required)):

Zhang et al present a single-cell analysis of acral melanomas and cutaneous melanoma from 4 and 2 patients, respectively, including one patient with acral melanoma with pre/post sample on PD1 therapy.

They propose the presence of melanoma subgroups with distinct representation in acral vs. cutaneous melanoma; different cell type proportions; increased abundance of Tregs in acral; and varying cell states among CD8 + T cells; they argue that T cells from acral are more likely to be exhausted. In a one-patient pre/post comparison they find pathways and inferred genomic alterations which they link to response outcomes. Treatment of acral melanoma is extremely challenging given the low response rates to immunotherapies and absence of druggable oncogenic drivers, as such, this is an important topic. However the study has serious limitations.

Response:

Thanks for your comments on this study. Given the reality of the low incidence of the acral subtype, efforts have been made to collect as many fresh samples as possible for single-cell sequencing analysis. In order to make up for the limitation of the number of our samples, we performed Bulk-RNA sequencing on 57 frozen tissue samples. At the same time, multiplex immunofluorescence staining and experiment of drug treatment at the cellular level were added to further improve the credibility of this study. We revised and polished our manuscript. Now we have 6 figures and 6 supplemental figures.

First, the sample sizes are simply too small. I recognize that collection of such samples is challenging, but no meaningful comparisons can be drawn from comparing 4 vs. 2 patients, wherein repeated sampling of the same patient (two in the acral group and one in the cutaneous group) is not even accounted for. The inability to apply proper statistics is pervasive in the entire manuscript and results in a lot of descriptions, such as "higher abundance of cell type X" without any statistical evaluation of these statements. When statistical evaluations are applied (e.g. abundance of Tregs, Fig. 3d) there is no correction for multiple hypothesis testing, which would be dismal in any of these cases due to low sample size (in fact, only n=2 in the cutaneous group)

Response:

Thanks for your comment. During the following six months, we did treat some acral melanoma patients, but we could not add single cell sequencing data because of lack high quality fresh tissues. We added an additional 57 melanoma samples subjected to RNA-seq, 6 samples to TCR-seq and 8 samples to whole exome sequencing. In addition, we added the confirmatory experiments of multiplex immunofluorescence staining for the identified markers, drug treatment for testing the efficacy of combinatorial anti-*PD1* and anti-*TIM3* treatment and comparative analysis against

public datasets etc. The findings derived at the single-cell level were well validated.

We also modified the statistical evaluation of **Figure 3D** to include multiple hypothesis testing. In order to further explore this result, we performed multiplex immunofluorescence staining for *CD4* and *FOXP3* on tissue sections at 5 AM and 3 CM. The results are shown in **Figure 3E-F**, which more intuitively showed that acral melanoma has more Treg infiltration. And we selected 5 representative fields of each section to count positive cells, and made statistical analysis between groups on the count results. The results showed that the immune infiltrate of acral melanoma was enriched with more Treg cells (**Figure 3F**).

Second, the clustering as shown clearly demonstrates that the data suffers from serious batch effects that are not accounted for. Furthermore, the boundaries for clusters (e.g. Fig 2a) seem arbitrary and suffer from serious batch effects (e.g. cluster 1 made up completely by one patient). Further evidence for batch effects are analyses presented in figure 4D, clearly showing that T cell populations are extremely variable for the exhausted score, indicating that this effect is driven by individual samples.

Response:

Thanks for your comments. We used the of *kBET* acceptance rate⁶ as a measurement of batch-effect. The acceptance rate measures whether cells from different batches are well-mixed in the local neighborhood of each cell. The acceptance rate obtained from our single-cell analytical pipeline Miscell⁷ (recently published in iScience <https://doi.org/10.1016/j.isci.2021.103200>) was comparable with the other batch-correction methods such as Seurat (v3.1.5)^{8,9}, Combat (v1.8.0)¹⁰, Scanorama (v1.7.1)¹¹, Harmony (v0.1.6)¹² and scVI (v0.0.0)¹³ (**Response Figure 4A-E**). This suggested that batch-effect was well addressed by our analytical method. Meanwhile, cluster 1 is consisted of cells from multiple patients (**Response Figure 4N**): 23% from Acral-1, 34% from Acral-2, 16% from Cutaneous-1, 55% from Cutaneous1-Lym. We examined the difference of exhausted score by dividing T cell populations into different segment according to their distribution. We observed that each segment was admixed from different patients (**Response Figure 4G, H and L**).

Response Figure 4. Benchmark of Miscell (our single-cell analytical pipeline) against other batch-correction methods. (A-E) The t-SNE plots of Seurat, Combat, Scanorama, Harmony and scVI. **(F)** Barplot showing the kBET acceptance rate among 6 different methods. **(G)** Violin plot and density plot showing the 3 segments of the exhausted score. **(H-L)** Barplot showing the ratio of 3 segment in each sample **(H)** and the ratio of samples in each phase **(L)**. **(M)** The t-SNE plot, showing cell origins by color, patient origin (right panel), and enlarging cluster1 cells. **(N)** Barplot showing the ratio of each sample in cluster1.

Third, the pre/post comparison using genomic inference is a bit overinterpreted. Only because cells with chr. 4 were not detected in the pre-treatment specimen, does not prove that they were not present. the tool they use to narrow down on gene expression (inferCNV) is furthermore not designed to define small gene segment windows, yet they use the "differential expression" of genes on Chr 4 in post vs pre to make further inferences.

Response:

Thanks a lot for your comment. We agree that inferCNV is not able to narrow down to gene expression. The amplification of Chr4 identified by inferCNV motivated us to test the differential expression of all genes located at Chr4 in single-cells from post-treatment sample versus pre-treatment sample, giving rise to 44 genes exhibited differential expression.

Overall, my strong recommendation is that the authors consider these weaknesses, increase the sample sizes when possible, or otherwise stick to a purely descriptive approach to interpreting this data.

Response:

Thank you for your honest suggestion. We added an additional 57 melanoma samples subjected to RNA-seq, 6 samples to TCR-seq and 8 samples to whole exome sequencing. In addition, we added the confirmatory experiments of multiplex immunofluorescence staining for the identified markers, drug treatment for testing the efficacy of combinatorial anti-*PDI* and anti-*TIM-3* treatment and comparative analysis against public datasets etc. The findings derived from single-cell sequencing were well validated.

Reference

1. Tirosh, I., *et al.* Dissecting the multicellular ecosystem of metastatic melanoma by single-cell RNA-seq. *Science (New York, N.Y.)* **352**, 189–196 (2016).
2. Sade-Feldman, M., *et al.* Defining T Cell States Associated with Response to Checkpoint Immunotherapy in Melanoma. *Cell* **175**, 998–1013.e1020 (2018).
3. Genomic Classification of Cutaneous Melanoma. *Cell* **161**, 1681–1696 (2015).
4. Stein, J.E., *et al.* Major pathologic response on biopsy (MPRbx) in patients with advanced melanoma treated with anti-PD-1: evidence for an early, on-therapy biomarker of response. *Annals of oncology : official journal of the European Society for Medical Oncology* **30**, 589–596 (2019).
5. Sun, Y., *et al.* Single-cell landscape of the ecosystem in early-relapse hepatocellular carcinoma. *Cell* **184**, 404–421.e416 (2021).
6. Buttner, M., Miao, Z., Wolf, F.A., Teichmann, S.A. & Theis, F.J. A test metric for assessing single-cell RNA-seq batch correction. *Nat Methods* **16**, 43–49 (2019).
7. Shen, H., *et al.* Miscell: An efficient self-supervised learning approach for dissecting single-cell transcriptome. *iScience* **24**, 103200 (2021).
8. Butler, A., Hoffman, P., Smibert, P., Papalexi, E. & Satija, R. Integrating single-cell transcriptomic data across different conditions, technologies, and species. *Nat Biotechnol* **36**, 411–420 (2018).
9. Kang, H.M., *et al.* Multiplexed droplet single-cell RNA-sequencing using natural genetic variation. *Nat Biotechnol* **36**, 89–94 (2018).
10. Johnson, W.E., Li, C. & Rabinovic, A. Adjusting batch effects in microarray expression data using empirical Bayes methods. *Biostatistics* **8**, 118–127 (2007).
11. Hie, B., Bryson, B. & Berger, B. Efficient integration of heterogeneous single-cell transcriptomes using Scanorama. *Nature biotechnology* **37**, 685–691 (2019).
12. Korsunsky, I., *et al.* Fast, sensitive and accurate integration of single-cell data with Harmony. *Nat Methods* **16**, 1289–1296 (2019).
13. Lopez, R., Regier, J., Cole, M.B., Jordan, M.I. & Yosef, N. Deep generative modeling for single-cell transcriptomics. *Nat Methods* **15**, 1053–1058 (2018).

Reviewers' Comments:

Reviewer #1:

Remarks to the Author:

The authors have addressed the principal comments in the review.

Reviewer #2:

Remarks to the Author:

The authors have done a tremendous job answering my concerns but there are just a few minor things the authors need to address. Other than these minor comments, the manuscript has improved significantly.

1. I am not an immunologist, but it is not clear from the immunofluorescent staining whether the checkpoint expression shown in Figure 5 is specific to CD8+ T cells. The staining for CD8 is not visible, and the bar graphs are only showing relative expression of each checkpoint marker but that doesn't mean these are expressed on T cells. Similarly, its hard to tell which cells are expressing GZMB. It just looks like the overall tissue in general has more TIM3 and PD1 in the acral samples and less GZMB (but doesn't seem to be obviously specific to CD8 T cells).

2. The datasets shown in supplemental Figure 6 which are intended to support/validate findings in Figure 5 on the changes in CD4 and CD8 cells pre/post immune therapy treatment are cutaneous melanoma data sets, where-as the original Figure 5 is focused on the pair of acral melanoma specimens. Not only do the authors not make this distinction very clear in the results section but its also not completely clear to this reviewer if this is an appropriate dataset to validate the original findings since original data was in acral melanoma and the validation data set is from a cutaneous melanoma sample set. At the very least this limitation should be explicitly mentioned in the manuscript and it should be clear in the results section that the two data sets are from different melanoma subtypes.

3. What is Supplemental Figure 3L showing? Are these T cells, or all cells? Please provide some kind of annotation on the new supplemental Figure 3L to show which samples the cells shown are coming from and on the key include the survival time so that it can be easily interpreted if the MT1E ad MT2A expression was higher on T cells from patients with worse prognosis.

4. Please include a comment in the discussion of the manuscript about the improved survival observed in acral melanoma patients vs cutaneous at their institution in the context of worse therapy responses observed in acral patients, since typically acral melanoma responses are worse and are characterized by an immune-cold environment.

Reviewer #3:

Remarks to the Author:

Zhang et al provide a revised manuscript describing acral melanomas using single-cell RNA-seq, now adding bulk RNA-seq, bulk TCR and WES on a subset.

While the authors have put considerable effort into this revision, they unfortunately did not address the critiques from the first submission.

It is still clear that the data suffers from serious batch effects. The fact that multiple methods for batch correction achieve similar results does not change that fact. The authors failed to address a key

critique, that is to increase samples size or attempt to apply proper statistics.

We are left with a similarly statistically poorly evaluated single-cell RNA-seq data set, but addition of bulk RNA_seq data. Unfortunately, I am not sure how meaningful the additional RNA-seq data and few WES and TCR profiling data sets are, as most of the knowledge gleaned from those analyses here have been demonstrated in larger data sets and in better controlled studies (e.g. Hayward et al., Nature, 2017). To no fault of the authors, but simply accepting the state of the field, there has been a publication of a previously presented study using single-cell RNA-seq of acral melanoma including 9 specimens in the meantime (Jiannong Li et al., Clinical Cancer Research, 2022).

As such, it appears that the current manuscript still requires further and improved statistical evaluation, ideally through increase in sample size, before meaningful and novel insights can be gleaned and robustly tested.

Response Letter to Reviewers

Comments from the Reviewers:

Reviewer #1 (Remarks to the Author):

The authors have addressed the principal comments in the review.

Response: Thank you for the support.

Reviewer #2 (Remarks to the Author):

The authors have done a tremendous job answering my concerns but there are just a few minor things the authors need to address. Other than these minor comments, the manuscript has improved significantly.

1. I am not an immunologist, but it is not clear from the immunofluorescent staining whether the checkpoint expression shown in Figure 5 is specific to CD8+ T cells. The staining for CD8 is not visible, and the bar graphs are only showing relative expression of each checkpoint marker but that doesn't mean these are expressed on T cells. Similarly, its hard to tell which cells are expressing GZMB. It just looks like the overall tissue in general has more TIM3 and PD1 in the acral samples and less GZMB (but doesn't seem to be obviously specific to CD8 T cells).

Response: Thank you for the encouragement and the questions. We have updated Figure 5 by merging the staining of CD8A (marker of CD8+ T cells) and PD1, TIM-3 and GZMB, respectively. The updated **Figure 5** showed that checkpoints such as PD1, TIM-3 and GZMB are expressed specifically on CD8+ T cells. The updated **Figure 5A and B** exhibited that CD8A and PD1, TIM-3 and GZMB are respectively superimposed. This suggested that PD1, TIM-3 and GZMB are expressed by CD8+ T cells. Indeed,

CD8+ T cells have more expression of TIM-3 and PD1 and less GZMB in acral samples.

2. The datasets shown in supplemental Figure 6 which are intended to support/validate findings in Figure 5 on the changes in CD4 and CD8 cells pre/post immune therapy treatment are cutaneous melanoma data sets, where-as the original Figure 5 is focused on the pair of acral melanoma specimens. Not only do the authors not make this distinction very clear in the results section but its also not completely clear to this reviewer if this is an appropriate dataset to validate the original findings since original data was in acral melanoma and the validation data set is from a cutaneous melanoma sample set. At the very least this limitation should be explicitly mentioned in the manuscript and it should be clear in the results section that the two data sets are from different melanoma subtypes.

Response: Thanks for your comments. We revised the Results section to make it clear. Specifically, we revised sentence in Lines 287 – 288 as: “In the Sade-Feldman’s cohort that consists of 32 patients diagnosed with cutaneous melanoma” and sentence in Lines 291 – 292 as: “In the other Bulk-RNA cohort consisted of 42 cutaneous melanoma patients that had both pre- and post-treatment samples”

We also added it as a limitation in the Discussion section in Lines 364 – 367: “However, the expression levels of these 44 genes in SD/PD group versus CR/PR group in acral melanoma patients remain to be investigated in future study given that RNA-seq data of acral melanoma patients receiving immunotherapy treatment are not yet available”

3. What is Supplemental Figure 3L showing? Are these T cells, or all cells? Please provide some kind of annotation on the new supplemental Figure 3L to show which samples the cells shown are coming from and on the key include the survival time so that it can be easily interpreted if the MT1E and MT2A expression was higher on T cells from patients with worse prognosis.

Response: Supplemental Figure 3L showed a specific cluster of CD8+ T cells, featured by high expression of *MT2A* and *MT1E*. We added survival data to Supplemental Figure 3L (updated as Figure S5 in the revised text).

4. Please include a comment in the discussion of the manuscript about the improved survival observed in acral melanoma patients vs cutaneous at their institution in the context of worse therapy responses observed in acral patients, since typically acral melanoma responses are worse and are characterized by an immune-cold environment.

Response: Thank you very much for your comments. We have added the following description in Discussion section in Lines 342 – 346 as: “We observed that Immune infiltration is scarce in patients with AM, which was also reported in previous study. However, AM has better overall survival as compared with CM. This is probably due to differences of tumor signatures underlying AM and CM. For instance, AM was enriched for cholesterol metabolism. Upregulation of the cholesterol metabolism was associated with favorable survival in lower grade glioma.”

Reviewer #3 (Remarks to the Author):

Zhang et al provide a revised manuscript describing acral melanomas using single-cell RNA-seq, now adding bulk RNA-seq, bulk TCR and WES on a subset.

While the authors have put considerable effort into this revision, they unfortunately did not address the critiques from the first submission.

It is still clear that the data suffers from serious batch effects. The fact that multiple methods for batch correction achieve similar results does not change that fact. The authors failed to address a key critique, that is to increase samples size or attempt to apply proper statistics.

We are left with a similarly statistically poorly evaluated single-cell RNA-seq data set, but addition of bulk RNA-seq data. Unfortunately, I am not sure how meaningful the additional RNA-seq data and few WES and TCR profiling data sets are, as most of the knowledge gleaned from those analyses here have been demonstrated in larger data sets and in better controlled studies (e.g. Hayward et al., Nature, 2017). To no fault of the authors, but simply accepting the state of the field, there has been a publication of a previously presented study using single-cell RNA-seq of acral melanoma including 9 specimens in the meantime (Jiannong Li et al., Clinical Cancer Research, 2022).

As such, it appears that the current manuscript still requires further and improved statistical evaluation, ideally through increase in sample size, before meaningful and novel insights can be gleaned and robustly tested.

Response: We thank the reviewer for the specific critiques and suggestions. We agree that the recent report by Jiannong Li et al provided us with a valuable external set for validation. In addition, we have accumulated additional samples from our cancer hospital to increase the sample size. We hope the reviewer would recognize our effort

in addressing important clinical issues for a rare cancer subtype.

Regarding the critique on the statistical methods and sample size, we have amended the statistics method in the evaluation of the mean difference and included additional samples to verify of key findings. We performed scRNA-seq for 2 acral and 1 cutaneous melanoma samples collected from Tianjin Cancer Hospital in the past six months and took it as an independent validation set. In addition, we downloaded scRNA-seq data of 9 acral melanoma samples from Jiannong Li et al and used it as external validation set. Cutaneous melanoma is not available from Jiannong Li et al. Additional results showed that the key findings such as different functional tumor cell subgroups (Signature1-5) and immune infiltration patterns deciphered from 8 samples (discovery set) reported in main text were verified. Detailed description is posted below.

Due to the violation of Gaussian distribution, we used Wilcoxon rank sum test instead of student's t-test to evaluate the difference of Tregs infiltration (**Figure 3F**), signature scores of CD8+ T cells (**Figure 4D**), proportion of CD8+ T cells expressing *PDI*, *TIM-3* and *GZMB* (**Figure 5C-E**) and infiltration of B cells, CD4+ and CD8+ T cells in AM3 sample (**Figure 6G**).

For the infiltration of T cells in AM versus CM, we constructed 2-by-2 contingency table (See below example) by randomly sampling one from the AM group and one sample from CM group. Subsequently, we calculated the odds ratio and applied Fisher's exact test to evaluate the difference of each immune cell cluster in the randomly selected AM sample versus CM sample. In total, we obtained 8 (i.e., $C_4^1 \times C_2^1$) contingency tables. We used the radar plot to depict the analytical results (**Figure 3D**).

An example of contingency table of Tregs constructed from AM1 and CM1:

	Treg	Non-Treg	Total
AM1	a	b	$a+b$

CM1	c	d	c+d
Total	a+c	b+d	a+b+c+d

Figure 3D. Radar plots depicting the odds ratio for AM versus CM sample.

We used Wilcoxon rank sum test to evaluate whether infiltrations of different immune cell clusters are different between AM versus CM samples. That is to test whether the median of odds ratio for each immune cell cluster in the above radar plot is different from 1. We observed that marginal higher infiltration of Tregs in AM as compared with CM (**Figure 3D and Table S6**, median odds ratio = 7.42, adjusted P-value=0.09). Infiltration of CD8-GZMK was lower in AM versus CM (**Figure 3D and Table S6** median odds ratio = 0.18, adjusted P-value=0.06).

We mapped 31974 cells from the internal validation set onto the tSNE plot of the discovery set and observed that cells from the discovery and internal validation sets are well mixed (**Figure S3A**, kBET = 0.846). We then updated the cell cluster identity by taking into account the inclusion of internal validation set (**Figure S3B**). Subsequently,

we picked up tumor cell clusters of the internal validation set (**Figure S3C**) and annotated the expression signatures of these tumor cell clusters. We observed signature patterns obtained from internal validation set (**Figure S3D-E**) are visually analogous to the signatures of discovery set (**Figure 2C**). Quantitatively, we observed high correlation of each corresponding signature in the discovery and internal validation sets (**Figure S3F**). In addition, functional signatures of the internal validation set are consistent with functional signatures derived from the discovery set. For instance, in the internal validation set, Signature 1 was involved in cholesterol transportation and phospholipid efflux; Signature 2 was enriched for Wnt signaling pathway and oxidative phosphorylation circuits; Signature 3 was featured by enrichment of Cell cycle circuits such as G2M checkpoint and E2F targets; Signature 4 was associated with TGF- β signaling and Signature 5 was enriched for interferon response. This suggested that tumor signatures identified in the discovery set was verified in the internal validation set.

We did the same analysis as above for the external validation set. We observed that cells from the discovery and external validation sets are also well mixed (**Figure S4A**, kBET = 0.827). We updated the cell cluster identity by taking into account the inclusion of external validation set (**Figure S4B**) and picked up tumor cell clusters from the external validation set (**Figure S4C**). We observed consistent signature patterns of tumor cell clusters obtained from the discovery set (**Figure 2C**) and external validation set (**Figure S4D-E**). Quantitatively, we observed high correlation of each corresponding signature in the discovery and external validation sets (**Figure S4F**). In external validation set, we observed that Signature 1 was involved in cholesterol transportation and phospholipid efflux. Signature 2 was enriched for Wnt signaling pathway and oxidative phosphorylation circuits. Signature 3 was featured by enrichment of Cell cycle circuits such as G2M checkpoint and E2F targets. Signature 4 was associated with TGF- β signaling. Signature 5 was enriched for interferon response. This suggested that tumor signatures identified in the discovery set was verified in the external validation set.

Additionally, we analyzed the expression signatures of immune cell clusters picked from the internal validation set (**Figure S6A**). The results showed that the proportion of Tregs in CM3 was significantly lower than in AM5 (Fisher's exact test, OR = 0.416, adjusted P-value = 1.246e-11) and AM6 (Fisher's exact test, OR = 0.393, adjusted P-value = 8.802e-12) (**Figure S6B**). The cytotoxicity score of CD8+ T cells in acral melanoma was significantly lower than that in cutaneous melanoma (**Figure S6C**, Median: 4.30 versus 2.17; Wilcoxon rank sum test, $p < 0.001$). Lineage trajectory inference of CD8+ T cells showed that the exhausted branch from acral melanoma are characterized by higher expression of *PD1* and *TIM-3* (**Figure S6D-F**). This result is consistent with the finding identified in the discovery set. We were not able to compare the immune infiltration in AM versus CM in the external validation set due to the unavailability of CM.

Taken together, the aforementioned results indicated that the key findings reported from the discovery set was verified independently in the internal and external validation sets. Batch effects are unavoidable and difficult to distinguish from biological differences. Given that the key findings are reproducible in both the internal and external validation sets, we hope that reviewer would agree with us that we have addressed the main critiques by expanded sample size both internally and externally.

Reference

1. Li, J., *et al.* Single-cell Characterization of the Cellular Landscape of Acral Melanoma Identifies Novel Targets for Immunotherapy. *Clinical cancer research : an official journal of the American Association for Cancer Research* **28**, 2131-2146 (2022).

Reviewers' Comments:

Reviewer #2:

Remarks to the Author:

The authors have sufficiently addressed my concerns, and the manuscript is now ready for publication.

Reviewer #3:

Remarks to the Author:

While I do not think that all my concerns with respect to power have been addressed, the authors have shown significant efforts with collection of few additional samples and analysis of a recently published data set.

If they sufficiently indicate the limitations of their data set in the discussion of the manuscript, I do believe the data presented in the manuscript will be of important value to the community.

I would also include a citation of a recent study in untreated melanoma (Biermann et al., Cell, 2022), because I think it will be helpful to the reader to consider analyzing these data together with the current data presented here.

Please do make sure that the processed AND raw data are available at the time of publication.

Response Letter

Reviewer #2 (Remarks to the Author):

The authors have sufficiently addressed my concerns, and the manuscript is now ready for publication.

Response: Thank you for your comments and support for this study.

Reviewer #3 (Remarks to the Author):

While I do not think that all my concerns with respect to power have been addressed, the authors have shown significant efforts with collection of few additional samples and analysis of a recently published data set.

If they sufficiently indicate the limitations of their data set in the discussion of the manuscript, I do believe the data presented in the manuscript will be of important value to the community.

I would also include a citation of a recent study in untreated melanoma (Biermann et al., Cell, 2022), because I think it will be helpful to the reader to consider analyzing these data together with the current data presented here.

Please do make sure that the processed AND raw data are available at the time of publication.

Response: Thanks for your comments. We have added it as a limitation and cited the study (Biermann et al., Cell, 2022) in the discussion section in manuscript. And the accessions of processed and raw data were listed in the Data available section.